# BRIDGING PERCEPTION AND REASONING: TOKEN REWEIGHTING FOR RLVR IN MULTIMODAL LLMS

## ABSTRACT

Extending Reinforcement Learning with Verifiable Rewards (RLVR) to multimodal large language models (MLLMs) faces a fundamental challenge: their responses inherently interleave **perception-related tokens**, which ground visual content, with **reasoning-related tokens**, which construct reasoning chains. These token types instantiate distinct yet interdependent capacities — *visual grounding and symbolic reasoning* — making isolated optimization insufficient. Through token-level empirical analysis, we demonstrate that optimizing either perception- or reasoning-only tokens consistently underperforms full optimization, underscoring their inherent coupling. To address this, we propose a *plug-and-play* **To**ken-**R**eweighting (**ToR**) strategy that explicitly models this interdependence by identifying critical tokens of both types and dynamically reweighting them during RLVR training. Applied on top of existing methods (*e.g.*, GRPO and DAPO), ToR delivers consistent performance gains across multiple multi-modal reasoning benchmarks, achieving state-of-the-art performance with both accurate visual grounding and coherent reasoning.

## 1 INTRODUCTION

Reinforcement Learning with Verifiable Rewards (RLVR) has substantially advanced the reasoning ability of large language models (LLMs) on complex tasks (Lambert et al., 2025; Shao et al., 2024; Guo et al., 2025; Yang et al., 2025a). Extending RLVR to multimodal large language models (MLLMs), however, is non-trivial: generated responses *interleave* tokens that ground visual content (**perception**) with tokens that drive symbolic inference (**reasoning**), as illustrated in Figure 1. Existing RLVR variants for MLLMs typically optimize these capabilities in isolation — either via chain-of-thought objectives for reasoning (Huang et al., 2025; Wei et al., 2025) or perception-oriented rewards and augmentations for perception (Wang et al., 2025e; Xiao et al., 2025; **?**) — leaving their interaction underexplored.

We hypothesize that this separated optimization is suboptimal because perception and reasoning are fundamentally interdependent at the token level. To empirically validate this claim, we conduct a controlled "selective optimization" study under Group Relative Policy Optimization (GRPO) (Shao et al., 2024). We identify reasoning-related tokens via high **next-token entropy** (following recent insights on reasoning forks (Wang et al., 2025c; Cheng et al., 2025)), and perception-related

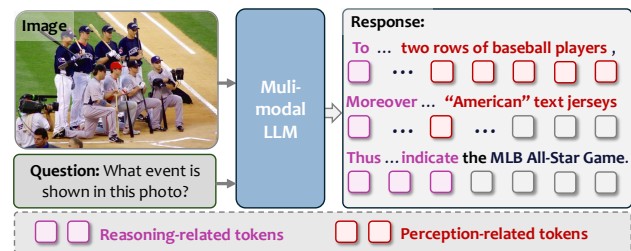

Figure 1: MLLM responses typically involve two types of critical tokens: **(1)** reasoning-related tokens to construct reasoning chains, and **(2)** perception-related tokens to ground and represent visual content.

tokens via **visual sensitivity**, measured as the change in token log-probability when conditioning on the image versus a text-only context (details in Section 3). We then train models while masking gradients on non-selected tokens, comparing three settings: optimizing only reasoning-related tokens, only perception-related tokens, and all tokens (vanilla GRPO).

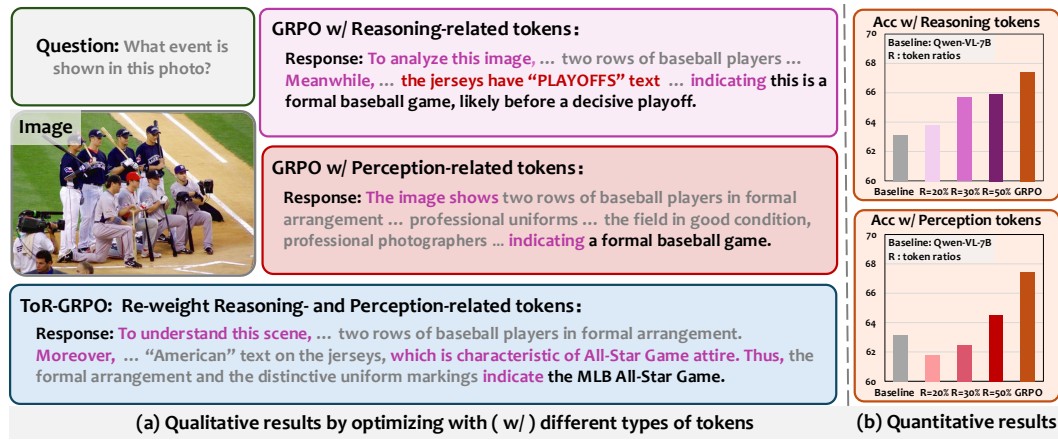

Figure 2: Performance comparison over the wemath benchmark (Qiao et al., 2024) when optimizing different token types with GRPO (Shao et al., 2024). Results across selection ratios 20%, 30%, or 50% show that optimizing either reasoning-only or perception-only tokens underperforms all tokens. Qualitative examples are selected from the best-performing checkpoints.

Across selection ratios (20%, 30%, 50%), optimizing only reasoning tokens underperforms vanilla GRPO (*e.g.*, ∼2% absolute drop on we-math benchmark), and optimizing only perception tokens fares worse (*e.g.*, ∼3% drop, with 20% and 30% tokens even worse than the baseline model); neither matches training on all tokens (Figure 2). Qualitatively, reasoning-only models produce coherent-looking chains of thought yet misinterpret key visual content, while perception-only models preserve low-level grounding but fail to integrate it into coherent reasoning. These results support our hypothesis: *perception and reasoning are coupled capabilities that demand joint optimization*.

Building on this insight, we propose **To**ken-**R**eweighting (**ToR**), a lightweight, plug-and-play module for RLVR that jointly optimizes perception and reasoning. Instead of treating all tokens equally or optimizing subsets in isolation, ToR strategically identifies the most critical perception- and reasoning-related tokens and adaptively reweights their importance in the policy gradient calculation. This mechanism explicitly models their interdependence, encouraging the MLLM to integrate visual grounding into its logical deliberations. As shown in Figure 2, applying ToR to GRPO (**ToR-GRPO**) not only recovers the performance lost from selective optimization but surpasses the standard GRPO baseline, confirming our effectiveness. Our contributions are threefold:

- We conduct the first systematic, token-level analysis to reveal the critical interdependence between perception and reasoning in MLLMs. Our controlled experiments quantitatively demonstrate that optimizing either capability in isolation is detrimental.
- We introduce **To**ken-**R**eweighting (**ToR**), a plug-and-play RLVR training strategy that explicitly models this interdependence by dynamically reweighting critical perception- and reasoning-related tokens during policy optimization.
- We empirically demonstrate that ToR delivers consistent and substantial gains when integrated with state-of-the-art RLVR algorithms (*e.g.*, GRPO and DAPO), setting a new state-of-the-art across a diverse suite of both multi-modal reasoning and perception benchmarks.

## 2 PRELIMINARIES

In this section, we revisit Reinforcement Learning with Verifiable Rewards (RLVR) procedure and representative RLVR optimization strategies for Multi-modal Large Language Models (MLLMs).

### 2.1 REINFORCEMENT LEARNING WITH VERIFIABLE REWARDS

Reinforcement Learning with Verifiable Rewards (RLVR) enhances multi-modal large language models (MLLMs) by aligning model outputs with verifiable answers. Given a multi-modal input with ground truth from a batch of B samples, $(\mathrm{I}^b, \mathrm{q}^b)) \in \{\mathrm{I}^b, \mathrm{q}^b\}_{b=1}^B$, and $y^b \in \{y^b\}_{b=1}^B$, where $\mathrm{I}^b$,

$q^b$, and $y^b$ denotes the image, question, and ground-truth respectively, the model $\pi_\theta$ generates output $o^b$ containing a reasoning process and a prediction. The prediction is enclosed in \boxed{.} while reasoning is delimited by <think>...</think>, enabling automated verification against the ground truth answers. RLVR employs a binary reward function $\mathbf{R}(\cdot)$ to determine whether the answer is correct by comparing the model output $o^b$ with ground truth $y^b$. The goal of RLVR is to maximize the reward function, formalized as:

$$\mathcal{J}_{\text{RLVR}}(\theta) = \max_\theta \mathbb{E}_{\{(\mathrm{I}^b, q^b) | y^b)\}_{b=1}^B} \mathbb{E}_{o^b \sim \pi_\theta(\cdot | \mathrm{I}^b, q^b)}[\mathbf{R}(o^b, y^b)]. \tag{1}$$

## 2.2 RLVR Optimization Algorithms

**Group Relative Policy Optimization (GRPO).** As a widely adopted RLVR optimization strategy, GRPO stabilizes training by computing advantages within response groups (Shao et al., 2024). Concretely, given a batch of samples $\{(\mathrm{I}^b, q^b) \mid y^b\}_{b=1}^B$, the GRPO objective is:

$$\mathcal{J}_{\text{GRPO}}(\theta) = \mathbb{E}_{\{(\mathrm{I}^b, q^b)\}_{b=1}^B} \mathbb{E}_{\{o_i^b\}_{i=1}^G \sim \pi_{\theta_{\text{old}}}(o^b | \mathrm{I}^b, q^b)} \left\{ \frac{1}{G} \sum_{i=1}^{G} \frac{1}{|o_i^b|} \sum_{t=1}^{|o_i^b|} \right.$$

$$\min\left[ \frac{\pi_\theta(o_{i,t}^b | \mathrm{I}^b, q^b, o_{i,<t}^b)}{\pi_{\theta_{\text{old}}}(o_{i,t}^b | \mathrm{I}^b, q^b, o_{i,<t}^b)} \hat{A}_{i,t}^b, \ \text{clip}\left( \frac{\pi_\theta(o_{i,t}^b | \mathrm{I}^b, q^b, o_{i,<t}^b)}{\pi_{\theta_{\text{old}}}(o_{i,t}^b | \mathrm{I}^b, q^b, o_{i,<t}^b)}, 1 - \epsilon, 1 + \epsilon \right) \hat{A}_{i,t}^b \right] - \beta \, \mathbb{D}_{KL}[\pi_\theta \| \pi_{\text{ref}}] \left. \right\}, \tag{2}$$

where $G$ is the rollout group size for each input, and the clip function restricts the importance ratio within $[1 - \epsilon, 1 + \epsilon]$. Moreover, the advantage can be formulated as:

$$\hat{A}_{i,t}^b = \frac{\mathbf{R}_i^b - \text{mean}(\mathbf{R}^b)}{\text{std}(\mathbf{R}^b)}, \ \text{where} \ \mathbf{R}_i^b = \mathbb{I}(\texttt{is\_equivalent}(o_i^b, y^b)), \tag{3}$$

where $\mathbb{I}(\cdot)$ is the indicator function, is_equivalent$(\cdot)$ extracts the predictions from the model output $o_i^b$ and compares with the ground truth $y^b$.

**Decoupled Clip and Dynamic Sampling Policy Optimization (DAPO)**. DAPO (Yu et al., 2025b) improves upon GRPO by removing KL regularization and introducing clip-higher, dynamic sampling, and token-level loss from the GRPO loss, achieving new state-of-the-art performance. Specifically, the DAPO objective for the batch of samples $\{(\mathrm{I}^b, q^b) \mid y^b\}_{b=1}^B$ can be formalized as:

$$\mathcal{J}_{\text{DAPO}}(\theta) = \mathbb{E}_{\{(\mathrm{I}^b, q^b)\}_{b=1}^B} \mathbb{E}_{\{o_i^b\}_{i=1}^G \sim \pi_{\theta_{\text{old}}}(o^b | \mathrm{I}^b, q^b)} \frac{1}{\sum_{i=1}^G |o_i|} \sum_{i=1}^{G} \sum_{t=1}^{|o_i^b|} \left\{ \right. \tag{4}$$

$$\min\left[ \frac{\pi_\theta(o_{i,t}^b | \mathrm{I}^b, q^b, o_{i,<t}^b)}{\pi_{\theta_{\text{old}}}(o_{i,t}^b | \mathrm{I}^b, q^b, o_{i,<t}^b)} \hat{A}_{i,t}^b, \ \text{clip}\left( \frac{\pi_\theta(o_{i,t}^b | \mathrm{I}^b, q^b, o_{i,<t}^b)}{\pi_{\theta_{\text{old}}}(o_{i,t}^b | \mathrm{I}^b, q^b, o_{i,<t}^b)}, 1 - \epsilon_{\text{low}}, 1 + \epsilon_{\text{high}} \right) \hat{A}_{i,t}^b \right] \left. \right\},$$

where $\epsilon_{\text{low}}$ and $\epsilon_{\text{high}}$ decouple the clipping thresholds for negative and positive advantages, respectively. In this work, we apply our token-reweighting strategy to both GRPO and DAPO, demonstrating its general applicability across various RLVR optimization strategies.

## 3 Approach

In this section, we elaborate on our token re-weighting strategy in detail. Firstly, we identify and analyze the effects of reasoning- and perception-related tokens; then, we illustrate our dynamic reweighting scheme based on the token types, and demonstrate the application to existing RLVR algorithms, such as GRPO and DAPO.

### 3.1 Token Identification

In this subsection, we present our token identification strategy and analyze the effects of emphasizing different types of tokens during GRPO optimization.

### 3.1.1 REASONING-RELATED TOKENS

**Identification of reasoning-related tokens.** Recent works (Wang et al., 2025c; Cheng et al., 2025) have shown that high-entropy tokens often correspond to critical "forking" points in reasoning chains, directly reflecting the model's *decision uncertainty*. Moreover, retaining gradients for only a small subset of such tokens has been shown to benefit optimization, highlighting their potential importance for reasoning.

Motivated by this, we identify reasoning-related tokens using an entropy-based criterion. Given the $i$-th generated response $\mathbf{o}_i^b = \{o_{i,1}^b, o_{i,2}^b \ldots, o_{i,L_i}^b\}$ conditioned on the image $\mathrm{I}^b$, and then question $\mathrm{q}^b$ from the input batch $\{(\mathrm{I}^b, \mathrm{q}^b)\}_{b=1}^B$, the prediction entropy at position $t$ is computed as:

$$H_{i,t}^b = -\sum_{v \in \mathcal{V}_{\text{top-}p}} \mathrm{P}_\theta(o_{i,t}^b = v \mid \mathbf{o}_{i,<t}^b, \mathrm{I}^b, \mathrm{q}^b) \cdot \log \mathrm{P}_\theta(o_{i,t}^b = v \mid \mathbf{o}_{i,<t}^b, \mathrm{I}_i^b, \mathrm{q}_i^b), \qquad (5)$$

where $\mathcal{V}_{\text{top-}p}$ denotes the set of vocabulary tokens within the top-$p$ cumulative probability ($p = 0.95$). This top-$p$ truncation avoids the influence of extremely low-probability tokens and is consistent with the rollout sampling process.

To aggregate across the rollout batch, we collect all token entropies into a set:

$$\mathcal{H} = \{H_{i,t}^b \mid b = 1, \ldots, B; \ i = 1, \ldots, G; \ t = 1, \ldots, L_i^b\}. \qquad (6)$$

The reasoning-token set is then defined by selecting the top-$\alpha_r$ fraction of tokens with the highest entropy across the batch:

$$\mathcal{T}_r = \{(b, i, t) \mid H_{i,t}^b \geq \mathrm{Percentile}_{1-\alpha_r}(\mathcal{H})\}, \qquad (7)$$

where $\alpha_r$ controls the fraction of selected tokens. Intuitively, these high-uncertainty positions correspond to pivotal points where reasoning chains are shaped.

**Influences of reasoning-related tokens.** We evaluate the practical effect of the reasoning-token set $\mathcal{T}_r$ by conducting GRPO training constrained to tokens in this set with $\alpha_r = 20\%, 30\%, 50\%$. As shown in Figure 3, all selection ratios underperform the baseline that optimizes over all tokens. Notably, the $30\%$ and $50\%$ settings converge to similar outcomes, both falling short of the full-token GRPO. These results indicate that while $\mathcal{T}_r$ captures critical decision points, optimizing solely on it fails to preserve sufficient contextual information — *particularly when errors originate from perception* — motivating the investigation of perception-related tokens.

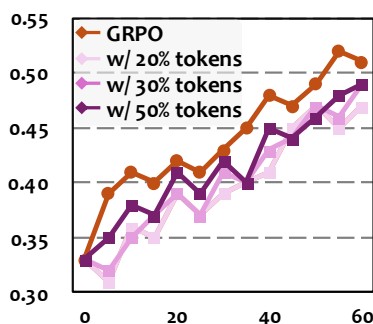

Figure 3: Performance on the Geo3K validation set for different ratios of reasoning-related tokens during GRPO training, Training data: Geo3K training set, Model: Qwen-VL-2.5 7B.

### 3.1.2 PERCEPTION-RELATED TOKENS

**Identification of perception-related tokens.** While reasoning-related tokens capture decision uncertainty, perception-related tokens highlight positions whose predictions strongly depend on visual inputs. To quantify this dependence, we compare the token log-probabilities under two conditions: (i) *with image*, conditioned on both $\mathrm{I}^b$ and $\mathrm{q}^b$; and (ii) *without image*, where the image channel is replaced by an empty placeholder $\varnothing$ and the model is conditioned only on $\mathrm{q}^b$.

Given the $i$-th generated response $\mathbf{o}_i^b = \{o_{i,1}^b, \ldots, o_{i,T_i}^b\}$, the visual-sensitivity score at position $t$ is computed as:

$$S_{i,t}^b = \big| \log \pi_\theta(o_{i,t}^b \mid \mathbf{o}_{i,<t}^b, \mathrm{I}_i^b, \mathrm{q}_i^b) - \log \pi_\theta(x_{i,t}^b \mid \mathbf{o}_{i,<t}^b, \varnothing, \mathbf{q}_i^b)\big|, \qquad (8)$$

where $\pi_\theta(o_t \mid \cdot)$ is the token-level policy. A large $S_{i,t}$ indicates a strong visual influence on token $o_{i,t}$. Aggregating across the rollout batch, the perception-token set is defined as the top-$\alpha_p$ fraction of tokens with the highest visual-sensitivity:

$$\mathcal{T}_p = \{(b, i, t) \mid S_{i,t}^b \geq \mathrm{Percentile}_{1-\alpha_p}(\{S_{b,i,t} \mid b = 1, \ldots, B; \ i = 1, \ldots, G; \ t = 1, \ldots, L_i^b\})\}, \qquad (9)$$

where $\alpha_p$ controls the selection ratio. This batch-level percentile selection ensures consistent token selection thresholds across rollout batches.

**Influences of perception-related tokens.** We evaluate the practical effect of the perception-token set $\mathcal{T}_p$ by constraining GRPO optimization to tokens in this set with $\alpha_p = 20\%, 30\%, 50\%$. As shown in Figure 4, all selection ratios underperform the full-token baseline. In particular, 20% and 30% result in performance drops exceeding 5%, whereas 50% shows the smallest performance gap but still underperforms the baseline. These results indicate that although perception-related tokens capture visually sensitive positions, optimizing solely on them is insufficient. *Effective GRPO training requires attending to both reasoning- and perception-related tokens to capture critical decision points while leveraging visual context.*

Figure 4: Performance on the Geo3K validation set for different ratios of perception-related tokens during GRPO training, Training data: Geo3K training set, Model: Qwen-VL-2.5 7B.

## 3.2 TOKEN REWEIGHTING

We unify reasoning- and perception-critical tokens into a single optimization framework by introducing *token reweighting*, which selectively amplifies their contributions while ignoring irrelevant tokens. Specifically, we integrate token-specific reweighting directly into the RL objectives. For the given input batch $\{(\mathrm{I}^b, \mathrm{q}^b)\}_{b=1}^B$, we constructed the reasoning-related token set $\mathcal{T}_r$ and the perception-related token set $\mathcal{T}_p$, and thus the GRPO objective with the token reweighting strategy (**ToR-GRPO**) is:

$$
\mathcal{J}_{\text{ToR-GRPO}}(\theta) = \mathbb{E}_{\{(\mathrm{I}^b, \mathrm{q}^b)\}_{b=1}^B} \mathbb{E}_{\{\boldsymbol{o}_i^b\}_{i=1}^G \sim \pi_{\theta_{\text{old}}}(\boldsymbol{o}^b | \mathrm{I}^b, \mathrm{q}^b)} \Bigg\{ \frac{1}{G} \sum_{i=1}^G \frac{1}{|o_i^b|} \sum_{t=1}^{|o_i^b|} \Big[ \boldsymbol{\gamma}_{\text{r}} \cdot \mathbb{I}[(\boldsymbol{b}, \boldsymbol{i}, \boldsymbol{t}) \in \boldsymbol{\mathcal{T}}_{\text{r}}]
$$

$$
+ \boldsymbol{\gamma}_{\text{p}} \cdot \mathbb{I}[(\boldsymbol{b}, \boldsymbol{i}, \boldsymbol{t}) \in \boldsymbol{\mathcal{T}}_{\text{p}}] \Big] \cdot \min \Big[ r_\theta(o_{i,t}^b) \hat{A}_{i,t}^b, \text{clip}\Big[ r_\theta(o_{i,t}^b), 1 - \epsilon, 1 + \epsilon \Big] \hat{A}_{i,t}^b \Big] - \beta \, \mathbb{D}_{KL}[\pi_\theta \, \| \, \pi_{\text{ref}}] \Bigg\}.
$$

$$(10)$$

Similarly, the DAPO objective with the token-weighting strategy (**ToR-DAPO**) is:

$$
\mathcal{J}_{\text{ToR-DAPO}}(\theta) = \mathbb{E}_{\{(\mathrm{I}^b, \mathrm{q}^b)\}_{b=1}^B} \mathbb{E}_{\{\boldsymbol{o}_i^b\}_{i=1}^G \sim \pi_{\theta_{\text{old}}}(\cdot | \mathrm{I}^b, \mathrm{q}^b)} \frac{1}{\sum_{i=1}^G |o_i^b|} \sum_{i=1}^G \sum_{t=1}^{|o_i|} \Bigg\{ \Big[ \boldsymbol{\gamma}_{\text{r}} \cdot \mathbb{I}[(\boldsymbol{b}, \boldsymbol{i}, \boldsymbol{t}) \in \boldsymbol{\mathcal{T}}_{\text{r}}]
$$

$$
+ \boldsymbol{\gamma}_{\text{p}} \cdot \mathbb{I}[(\boldsymbol{b}, \boldsymbol{i}, \boldsymbol{t}) \in \boldsymbol{\mathcal{T}}_{\text{p}}] \Big] \cdot \min \Big[ r_\theta(o_{i,t}^b) \cdot \hat{A}_{i,t}^b, \text{clip}\Big[ r_\theta(o_{i,t}^b), 1 - \epsilon_{\text{low}}, 1 + \epsilon_{\text{high}} \Big] \cdot \hat{A}_{i,t}^b \Big] \Bigg\}. \quad (11)
$$

$r_\theta(o_{i,t}^b)$ is the importance sampling ratio, formalized as:

$$
r_\theta(o_{i,t}^b) = \frac{\pi_\theta(o_{i,t}^b \mid \mathrm{I}_i^b, \mathrm{q}_i^b, o_{i,<t}^b)}{\pi_{\theta_{\text{old}}}(o_{i,t}^b \mid \mathrm{I}_i^b, \mathrm{q}_i^b, o_{i,<t}^b)}. \quad (12)
$$

Here, $\gamma_r$ weights reasoning-related tokens, emphasizing critical decision points in reasoning chains; $\gamma_p$ weights perception-related tokens, emphasizing the integration of visual context. Tokens outside these sets are excluded from optimization (*i.e.*, assigned a weight of zero). This reweighting ensures gradients focus on tokens essential for both reasoning and perception, improving both training efficiency and effectiveness.

**Discussion** Compared with existing approaches, ToR offers several key advantages: ❶ **Plug-and-play**. A simple reweighting mask integrates seamlessly into standard RLVR objectives, without

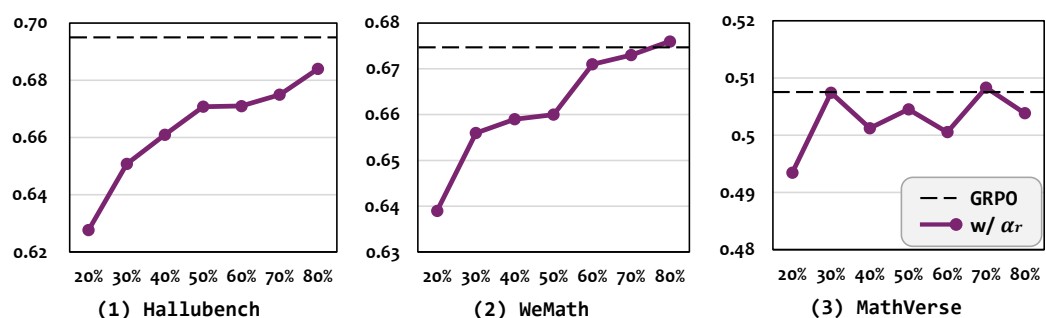

Figure 5: Performance comparison with different ratios of reasoning-related tokens ($\alpha_r$).

introducing extra pipeline modifications. ❷ **Self-contained.** Critical tokens are identified purely from the model's intrinsic uncertainty and visual sensitivity, eliminating the need for external priors. ❸ **Joint optimization.** By explicitly modeling the interdependence between reasoning and perception tokens, ToR enables balanced and simultaneous enhancement of both capabilities.

# 4 EXPERIMENT

In this section, we elaborate on the effectiveness of our token reweighting strategy. Specifically, we first introduce the details of our experimental settings. Next, we present the ablation studies, and finally, we compare our results with those of state-of-the-art methods across various benchmarks.

## 4.1 EXPERIMENTAL SETTINGS

We adopt the Geometry3K (Lu et al., 2021) dataset for training and validation, following existing methods (?Xiao et al., 2025), which consists of 2,100 training samples and 300 validation samples. Moreover, we employ the multi-modal framework EasyR1 (Yaowei et al., 2025) for reinforcement learning training, and following the works in (?) for evaluation. Evaluation is conducted with five benchmarks, including four for visual reasoning: MathVerse (Zhang et al., 2024), MathVision (Wang et al., 2024), MathVista (Lu et al., 2024), and WeMath (Qiao et al., 2024), and one for visual perception: HallusionBench (Guan et al., 2024).

**Implementation details.** We adopt Qwen2.5-VL-7B (Bai et al., 2025) as our baseline model by following existing works in (Meng et al., 2025; ?). Specifically, all experiments are conducted using 8 NVIDIA H800 GPUs (80 GB memory for each), with the default settings in EasyR1: a learning rate of $1e^{-6}$, a global batch size of 128, a rollout batch size of 512, a rollout $n$ as 12, and a rollout temperature of 0.95.

## 4.2 ABLATION STUDIES

In this section, we analyze the influences of different components on our token re-weighting strategy. Specifically, we utilize the Qwen-2.5-VL 7B (Bai et al., 2025) as the backbone, and conduct GRPO on the training set of Geometry3K. With the aim to evaluate the reasoning and perception ability, we select the Hallusion Bench (Guan et al., 2024), the Wemath (Qiao et al., 2024), and the MathVerse (Zhang et al., 2024) for evaluation, where Hallusion Bench focuses on the perception abilities, and MathVerse targets reasoning capabilities, whereas Wemath focuses on both.

❶ **The effects of different ratios for reasoning and perception tokens.** In this section, we study the effect of varying token ratios (*e.g.*, the proportion of reasoning-related tokens $\alpha_r$ and perception-related tokens $\alpha_p$). Results for reasoning-related tokens are presented in Figure 5, while those for perception-related tokens are shown in Figure 6. Specifically, we vary $\alpha_r$ and $\alpha_p$ from 20% to 80% over 60 GRPO steps, and report the performance of full-token GRPO as a dashed reference line.

From Figure 5, two observations emerge: (1) On perception-intensive benchmarks such as HallusionBench, none of the ratios matches the performance of full-token GRPO. (2) On reasoning-

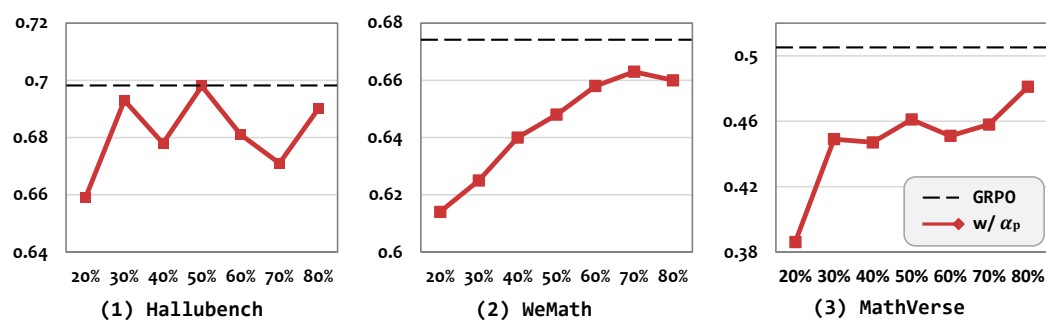

Figure 6: Performance comparison with different ratios of perception-related tokens ($\alpha_p$).

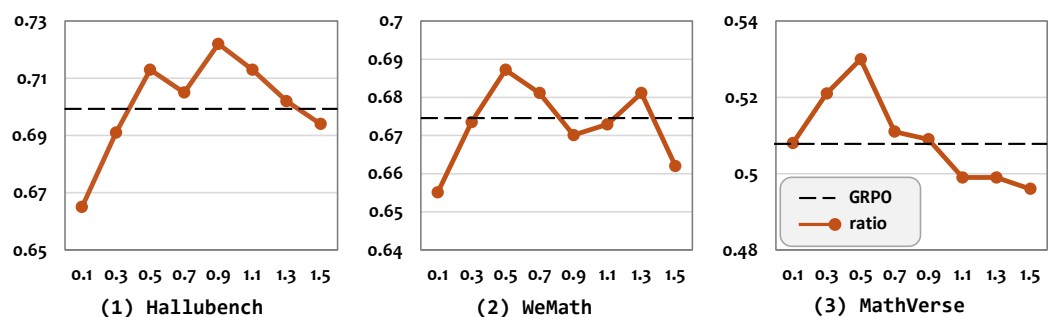

Figure 7: Performance comparison with varying combination ratios of reasoning- and perception-related tokens (reasoning weight $\gamma_r$ fixed at 1, perception weight $\gamma_p$ varying from 0.1 to 1.5).

focused benchmarks (*e.g.*, WeMath and MathVerse), adjusting token ratios produces results comparable to direct GRPO, but yields little to no further improvement.

Similarly, Figure 6 shows that while perception-only reweighting achieves performance close to full-token GRPO on perception-oriented benchmarks, it performs substantially worse on reasoning-demanding tasks, particularly on MathVerse, indicating a notable gap from full-token GRPO. These results collectively confirm that focusing exclusively on either reasoning- or perception-related tokens is insufficient for robust multimodal reasoning.

By jointly considering both token types and their dynamics during training, we adopt a balanced ratio of 30% reasoning tokens and 30% perception tokens for re-weighting. Notably, in the initial stage of GRPO training on Geometry3K, about 12% of tokens overlap between the two categories, and this proportion evolves during optimization. For these overlapping tokens, our experiments show that while both reasoning- and perception-related tokens experience performance drops, the decrease is smaller for reasoning tokens; therefore, we assign them the weight of reasoning tokens during re-weighting.

❷ **The effects of different combination ratios for different tokens.** In this section, we evaluate the impact of varying combination ratios between reasoning- and perception-related tokens. The results are shown in Figure 7, where the weight of reasoning tokens is fixed at 1, and the weight of perception tokens is varied from 0.1 to 1.5 in steps of 0.2. The performance of full-token GRPO is shown in a dashed reference line.

From Figure 7, we make the following observations: (1) Combining reasoning- and perception-related tokens consistently improves performance compared to using either type alone. (2) For reasoning-focused tasks, a relatively low proportion of perception tokens is preferable, whereas perception-oriented tasks benefit from a higher proportion. (3) Extremely low or high perception token weights tend to degrade performance. Overall, a weight of 0.5 for perception tokens provides a good balance across tasks, therefore, we adopt this ratio as a reference for subsequent experiments.

Table 1: Performance comparison of 7B-sized Multi-modal LLMs on different benchmarks. Following existing works in (**?**Xiao et al., 2025), we highlight the data size with blue and red for SFT and RL, respectively. The best value in each column is shown in **bold**, and the second-best is underlined.

| Model | Data Size | MathVerse | MathVision | MathVista | WeMath | HallusionBench |
|---|---|---|---|---|---|---|
| *Open-source Models* | | | | | | |
| InternVL-2.5-8B (Chen et al., 2024) | - | 39.5 | 19.7 | 64.4 | - | 67.3 |
| InternVL-3-8B (Zhu et al., 2025) | - | 39.5 | 29.3 | 71.6 | 37.1 | - |
| LLaVA-OneVision-7B (Li et al., 2025a) | - | 26.2 | - | 63.2 | - | 48.4 |
| Qwen2.5-VL-7B-Instruct (Bai et al., 2025) | - | 46.2 | 25.0 | 67.5 | 63.1 | 64.6 |
| *reinforcement learning with verifiable reward strategies* | | | | | | |
| R1-VL-7B (Zhang et al., 2025a) | 260K+10K | 40.0 | 24.7 | 63.5 | - | - |
| Vision-R1-7B (Huang et al., 2025) | 200K+10K | 52.4 | - | **73.5** | - | - |
| R1-OneVision-7B (Yang et al., 2025b) | 155K+10K | 46.1 | 22.5 | 63.9 | 62.1 | 65.6 |
| OpenVLThinker-7B (Deng et al., 2025b) | 35K+15K | 48.0 | 25.0 | 71.5 | 67.8 | 70.8 |
| MM-Eureka-Qwen-7B (Meng et al., 2025) | 15K | 50.5 | 28.3 | 71.5 | 65.5 | 68.3 |
| ThinkLite-7B-VL (Wang et al., 2025d) | 11K | 50.2 | 27.6 | 72.7 | 69.2 | 71.0 |
| VLAA-Thinker-7B (Chen et al., 2025a) | 25K | 49.9 | 26.9 | 68.8 | 67.9 | 68.6 |
| NoisyRollout-7B (Liu et al., 2025a) | 2.1K | 53.2 | 28.5 | 72.6 | 69.6 | 72.1 |
| GRPO (Shao et al., 2024) | 2.1K (Geometry3K) | 50.8 | 27.3 | 70.5 | 67.4 | 69.8 |
| **ToR-GRPO** | 2.1K (Geometry3K) | 53.0 | **28.6** | 71.9 | 68.9 | **72.4** |
| DAPO (Yu et al., 2025b) | 2.1K (Geometry3K) | 50.6 | 26.5 | 70.3 | 69.3 | 67.9 |
| **ToR-DAPO** | 2.1K (Geometry3K) | **53.4** | 27.9 | 72.6 | **72.1** | 71.8 |

## 4.3 COMPARISON WITH STATE-OF-THE-ART APPROACHES

In this section, we compare our token re-weighting strategies with state-of-the-art approaches, including open-source multi-modal LLMs: InternVL-2.5 (Chen et al., 2024), InternVL-3 (Chen et al., 2024), LLaVA-OneVision (Li et al., 2025a), and Qwen-2.5-VL 7B (Bai et al., 2025), as well as RLVR-based methods: R1-VL (Zhang et al., 2025a), Vision-R1 (Huang et al., 2025), R1-OneVision (Yang et al., 2025b), Open-VLThinker (Deng et al., 2025b), MM-Eureka (Meng et al., 2025), ThinkLite (Wang et al., 2025d), VLAA-Thinker (Chen et al., 2025a), and NoisyRollout (**?**). The results are summarized in Table 1.

Unlike existing approaches that often rely on large-scale data augmentation or extensive chain-of-thought distillation, our token re-weighting method achieves substantial improvements with only 2.1K samples. Specifically, it yields an average gain of more than 1.5% over GRPO across benchmarks, with notable improvements of 2.2% on MathVerse and 2.6% on HallusionBench.

Furthermore, our approach establishes new state-of-the-art results on multiple benchmarks. For example, on the WeMath benchmark, applying token re-weighting to DAPO (**ToR-DAPO**) achieves an improvement of about 3%, significantly surpassing state-of-the-art methods.

## 5 RELATED WORK

In this section, we first briefly summarize methods that focus on enhancing reasoning capabilities in Multi-modal LLMs, and then, we illustrate related methods for reinforcement learning with verifiable rewards. Finally, we enumerate the differences between our approach and related methods.

### 5.1 REASONING IN MULTI-MODAL LLMS

Existing approaches that focus on reasoning in multi-modal LLMs can be broadly categorized into:

❶ **Extending reasoning LLMs with visual understanding.** Building upon the strong reasoning capabilities of recent LLMs, one research branch explores incorporating visual content into LLMs. Typical strategies include: **(1)** integrating visual encoders into LLMs to directly extend them to multimodal scenarios (Peng et al., 2025; Wang et al., 2025b); and **(2)** transforming images into captions and feeding them into LLMs, enhancing chain-of-thought generation to bridge perception and reasoning (Huang et al., 2025; Wang et al., 2025e).

❷ **Enhancing reasoning abilities within MLLMs.** Another research branch seeks to endow existing MLLMs with reasoning skills. Representative approaches include: **(1)** transferring reasoning

priors from reasoning LLMs into MLLMs through model merging, thereby leveraging the complementary strengths of both models (Chen et al., 2025b); and **(2)** adapting RLVR algorithms, such as Group Relative Policy Optimization (GRPO) (Shao et al., 2024), to enhance reasoning capabilities in multimodal settings (Huang et al., 2025; Liu et al., 2025c).

## 5.2 REINFORCEMENT LEARNING WITH VERIFIABLE REWARDS

RLVR has recently demonstrated its effectiveness in aligning LLMs with verifiable reasoning outcomes. Among the implementations, GRPO (Shao et al., 2024) has shown great success with more stable advantage estimation. Subsequent refinements have further improved its efficiency and effectiveness (Yu et al., 2025b; Liu et al., 2025b; Wang et al., 2025c). When extending RLVR to multimodal settings, current methods primarily focus on two distinct branches:

❶ **Emphasizing the importance of visual understanding within RLVR.** Specifically, works like (Xiao et al., 2025; Yu et al., 2025a) introduce perception-oriented rewards to incentivize accurate visual grounding and understanding. Moreover, works such as (**?**Wang et al., 2025e; Li et al., 2025b) employ data augmentation strategies to improve model robustness and sensitivity against visual variations. Works like (Zhang et al., 2025c; Liu et al., 2025d) incorporate image manipulation tools like cropping and zooming in to focus on critical regions in the image during the reasoning process.

❷ **Constructing coherent reasoning chains with RLVR.** Representative works include: **(1)** distilling chain-of-thought reasoning patterns from stronger reasoning models to improve reasoning coherence in multimodal tasks (Huang et al., 2025; Wei et al., 2025); and **(2)** modifying different components of GRPO (e.g., clip ratios or advantage estimation) to emphasize reasoning-critical tokens better and stabilize training (Zhang et al., 2025b; Meng et al., 2025; Wang et al., 2025a).

**Differences.** Unlike existing methods that focus on optimizing either perception or reasoning abilities in isolation, our work systematically investigates and addresses the interdependence between these two capabilities. Specifically, through comprehensive token-level analysis, we demonstrate that perception and reasoning tokens exhibit complex interactions during training, where optimizing one type can inadvertently impair the other. To address this challenge, we propose a simple token-reweighting strategy that explicitly balances the optimization of both perception and reasoning tokens, leading to significant performance improvements across both capabilities.

## 6 CONCLUSION

In this work, through systematic token-wise analysis, we uncover a fundamental challenge in extending RLVR to multimodal LLMs: **the intrinsic interdependence between perception and reasoning.** We show that overemphasizing either capability inevitably impairs the other, yet current approaches overlook this issue and optimize them in isolation.

To address this, we proposed a simple yet effective **To**ken-**R**eweighting (**ToR**) strategy that identifies and reweights perception- and reasoning-related tokens during RLVR training. We apply ToR over current RLVR algorithms (*e.g.*, GRPO and DAPO), and achieves significant performance gains across diverse benchmarks, consistently enhancing both perception and reasoning.

**Limitations and Future Work.** While ToR establishes a foundation for addressing perception-reasoning interdependence, several promising directions remain open: ❶ **Fine-grained token identification strategies**: Precisely localizing critical regions in images using models like SAM (Kirillov et al., 2023) to identify more fine-grained perception-critical tokens. ❷ **Dynamic token reweighting**: Dynamically assigning weights to tokens based on their gradient contributions or connections to final outcomes (Yu et al., 2025c). ❸ **Extending beyond tokens**: As tokens derive meaning from context, future work could explore perception-reasoning interdependence at broader contextual levels — optimizing tokens within their semantic context, preserving their contextual relationships. ❹ **Exploring broader applications**: Extending this interdependence framework to more complex scenarios, *e.g.*, unified multi-modal generation and understanding tasks (Wu et al., 2025; Deng et al., 2025a), where visual tokens participate in reasoning processes.

ETHICS STATEMENT

Our token re-weighting (ToR) approach significantly enhances the reasoning capabilities of multi-modal LLMs. As a simple, plug-and-play strategy, it can be readily integrated into existing RLVR frameworks, making it a practical tool for improving multi-modal reasoning performance. Given its focus on model optimization, this method does not introduce new ethical risks beyond those already inherent to large language models.

REPRODUCIBILITY STATEMENT

Our ToR strategy is easy to implement. We provide implementation details, including the framework, token selection criteria, and token weights used in our experiments. The code will be made publicly available upon acceptance to facilitate reproducibility and further research.

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

## A    USAGE OF LLMS

We employed large language models (LLMs) solely for polishing the language of this paper. All content was originally drafted by the authors, and the use of LLMs was limited to refining pre-organized text and paragraphs. All suggested modifications were carefully reviewed by the authors to ensure accuracy and consistency with the intended meaning.

## B    TOKEN DISTRIBUTION AND CHARACTERISTIC ANALYSIS

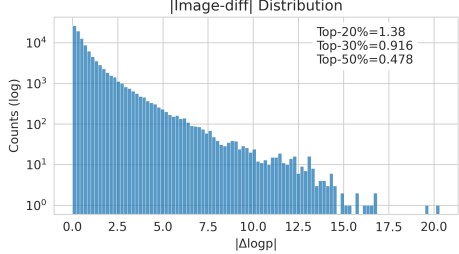 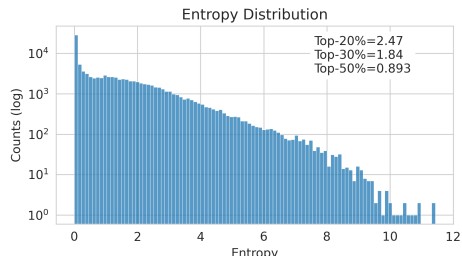

Figure 8: Distribution of log-probability differences for Qwen-2.5-VL-7B on Hallusion-Bench, used to identify perception-related tokens (Guan et al., 2024).

Figure 9: Distribution of entropy values for Qwen-2.5-VL-7B on HallusionBench, used to identify reasoning-related tokens (Guan et al., 2024).

To further investigate the characteristics of perception- and reasoning-related tokens, we analyze the token distribution of Qwen-2.5-VL-7B on both perception and reasoning benchmarks. Specifically, we report the entropy distribution (Figures 9, 11) and the image log-probability difference distribution (Figures 8, 10) on *HallusionBench* (Guan et al., 2024) and *WeMath* (Qiao et al., 2024), respectively. For each case, we further highlight the values corresponding to the top 20%, 30%, and 50% of tokens, ranked by their entropy or log-probability differences.

Our observations are as follows:

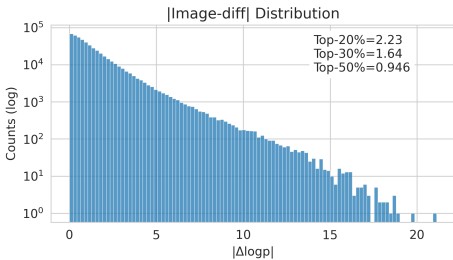 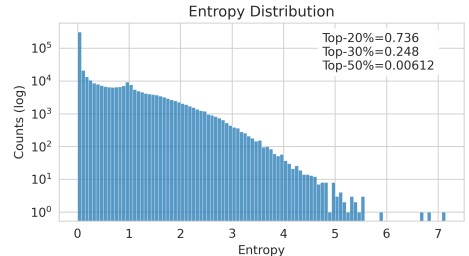

Figure 10: Distribution of log-probability differences for Qwen-2.5-VL-7B on We-Math, used to identify perception-related tokens (Qiao et al., 2024).

Figure 11: Distribution of entropy values for Qwen-2.5-VL-7B on WeMath, used to identify reasoning-related tokens (Qiao et al., 2024).

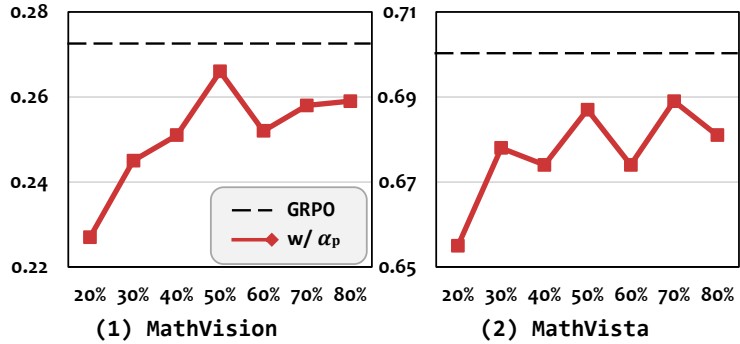

Figure 12: Additional experimental results with different ratios of perception-related tokens ($\alpha_p$) on *MathVision* and *MathVista*.

❶ Perception-related benchmark (Guan et al., 2024). High-entropy tokens exhibit large variations, with more than 50% of tokens exceeding $0.89$. In contrast, the image log-probability difference of perception-related tokens is relatively stable: more than 70% of tokens are less than $0.916$. This indicates that a small set of perception-related tokens is consistently stable and plays a critical role in capturing visual cues, suggesting that focusing on these tokens effectively enhances the model's perceptual ability.

❷ Reasoning-related benchmark (Qiao et al., 2024). In reasoning tasks, the distribution shows the opposite trend. Perception-related tokens present large variations, with more than 50% exceeding $0.946$. Meanwhile, high-entropy tokens remain relatively stable, with over 70% less than $0.248$. This implies that a small subset of reasoning-related tokens is more robust and can effectively capture the key reasoning process, highlighting their importance in enhancing the model's reasoning capability.

These results demonstrate that perception and reasoning rely on different types of stable tokens: perception emphasizes stability in a small number of visually sensitive tokens, while reasoning relies on the robustness of high-entropy tokens. This contrast validates our token re-weighting strategy that explicitly leverages both perception- and reasoning-related tokens.

## C ADDITIONAL EXPERIMENTAL RESULTS

In this section, we provide additional ablation results on two reasoning-focused benchmarks, Math-Vision (Wang et al., 2024) and MathVista (Lu et al., 2024). Specifically, we evaluate the impact of different ratios of perception-related tokens ($\alpha_p$), reasoning-related tokens ($\alpha_r$), and their combinations. The results are illustrated in Figure 12, Figure 13, and Figure 14, respectively.

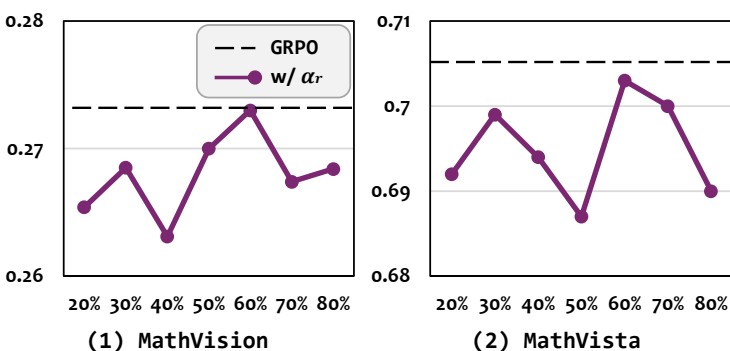

Figure 13: Additional experimental results with different ratios of reasoning-related tokens ($\alpha_r$) on *MathVision* and *MathVista*.

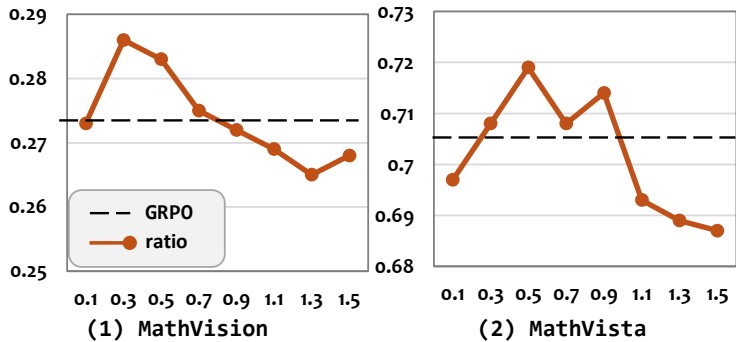

Figure 14: Additional experimental results with combined ratios of perception- and reasoning-related tokens ($\gamma_p = 1, \gamma_r$ from 0.1 to 1.5) on *MathVision* and *MathVista*.

From Figure 12 and Figure 13, we can observe that relying solely on perception- or reasoning-related tokens underperforms the full GRPO optimization, while focusing on reasoning tokens achieves relatively better results than perception tokens. Moreover, from Figure 14, the re-weighting strategy consistently improves performance, and the effectiveness of using 0.5 perception tokens across both benchmarks further validates the robustness of our re-weighting strategy.

## D  PERCEPTION-REASONING INTERDEPENDENCE

In this section, we demonstrate the interdependence between perception and reasoning tokens. Specifically, we visualize the relationship between reasoning uncertainty and perception strength over the training and validation set of Geo3K (Figure 15 & Figure 16), where we can observe a strong push-pull dynamic relationship between the reasoning and perception. Moreover, for clear illustration, we illustrate the relationship in Figure 17, and the learning dynamics as in Figure 18.

## E  PERCEPTION TOKEN IDENTIFICATION WITH VARIOUS CRITERIA

In this section, we compare different perception tokens selection criteria, including: "logp-diff", "probs-diff", "entropy-diff", and "attention scores" to the image, Experimental results across various datasets are listed in Figure 19 → Figure 27, where we can observe that "logp-diff" better trades how much the image matter and how meaningful is the change.

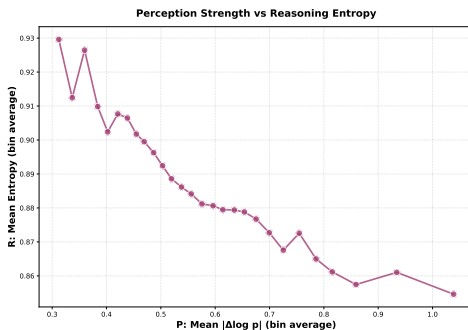

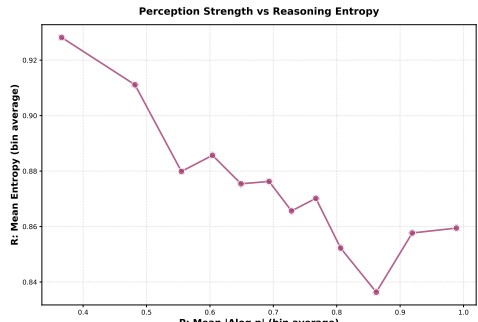

Figure 15: Relationship between (1) reasoning uncertainty: the entropy value of reasoning tokens and (2) perceptron strength: the logp diff between perception tokens for each response over Geo3K training set, response sampled from Qwen-2.5-VL 7B.

Figure 16: Relationship between (1) reasoning uncertainty: the entropy value of reasoning tokens and (2) perceptron strength: the logp diff between perception tokens for each response over Geo3K validation set, response sampled from Qwen-2.5-VL 7B.

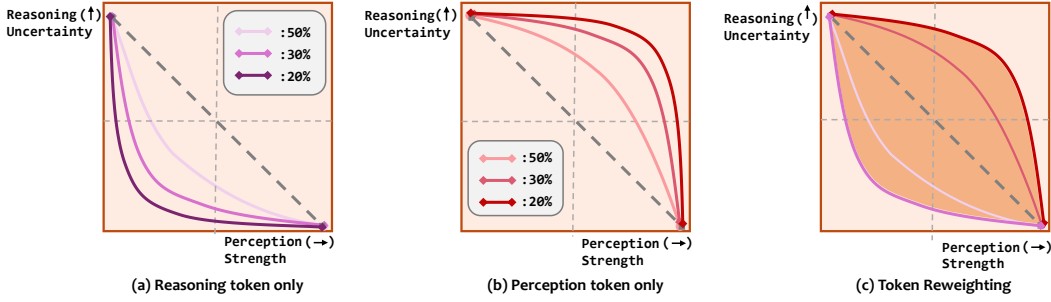

Figure 17: The comparison of perception strengths and reasoning uncertainty with different token ratios for GRPO optimization.

## F DISTRIBUTION OF SELECTED TOKENS OVER VARIOUS ROLLOUTS.

In this section, we show the distribution of selected tokens over a batch of rollouts as in Figure 28. We can find that different rollouts receive a comparable overall amount of optimization, but with different mixtures of tokens: harder groups are optimized more on reasoning, easier groups more on perception, while both token types remain well represented across the rollout batch.

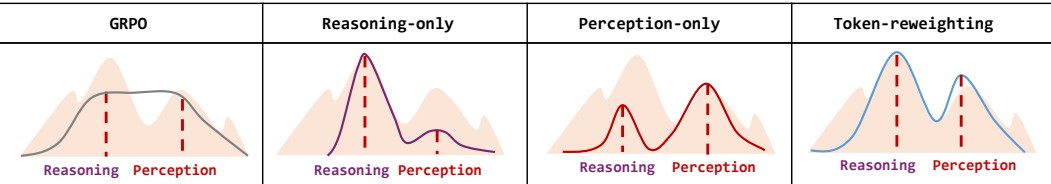

Figure 18: Comparison between vanilla GRPO, GRPO with reasoning-tokens only, GRPO with perception-tokens only, and GRPO with Token-reweighting.

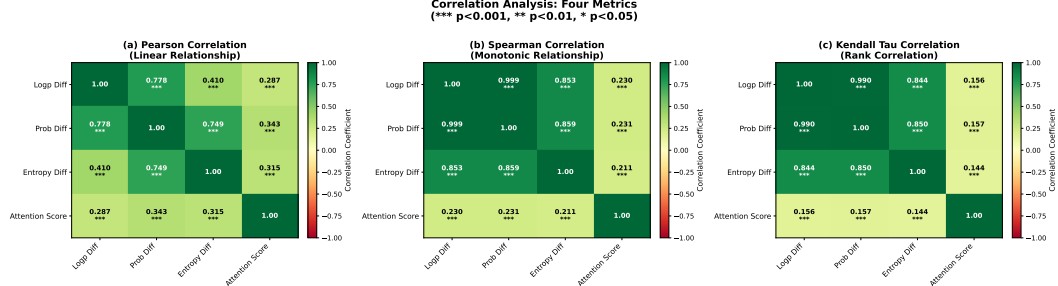

Figure 19: Correlation heatmaps between different image token selection strategies over the validation set.

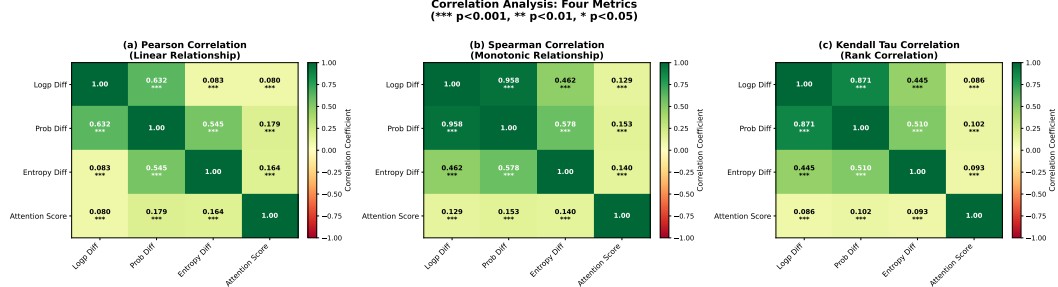

Figure 20: Correlation heatmaps between different image token selection strategies over the hallubench benchmark.

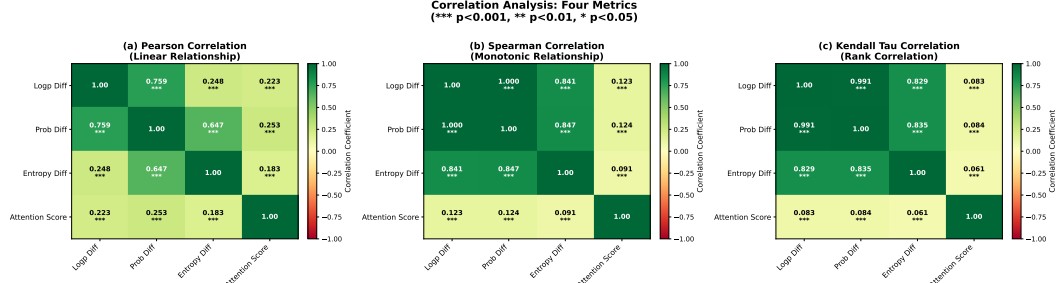

Figure 21: Correlation heatmaps between different image token selection strategies over the wemath benchmark.

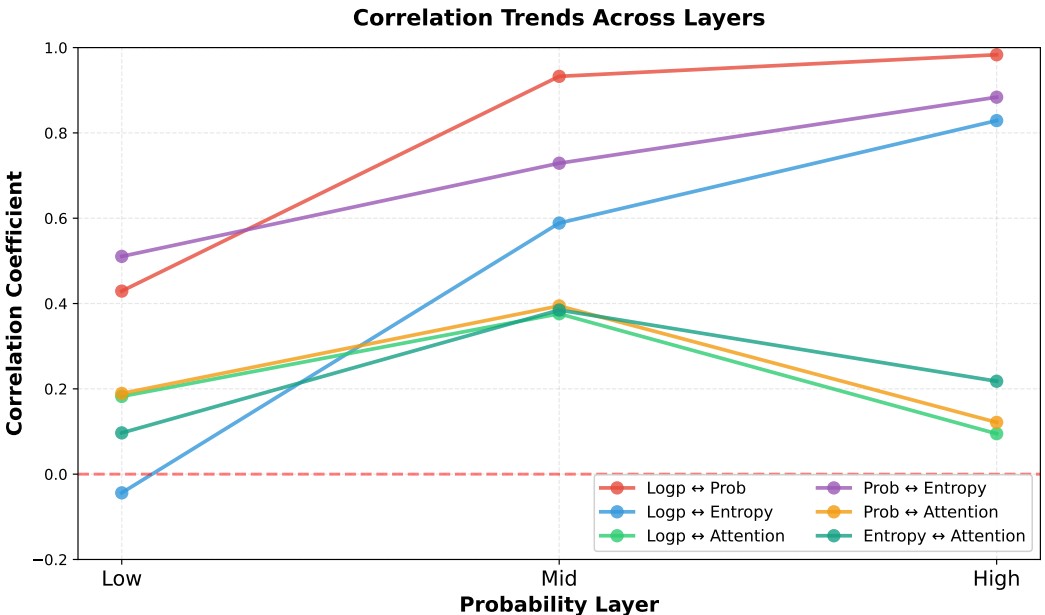

Figure 22: Correlation comparison across layers between different image token selection strategies over the validation set.

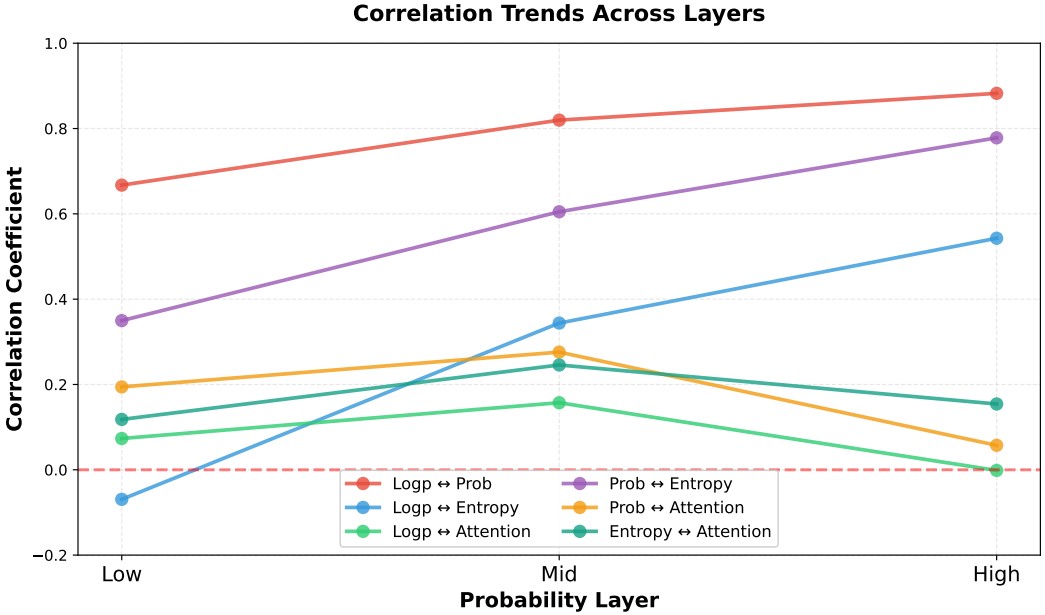

Figure 23: Correlation comparison across layers between different image token selection strategies over the hallubench benchmark.

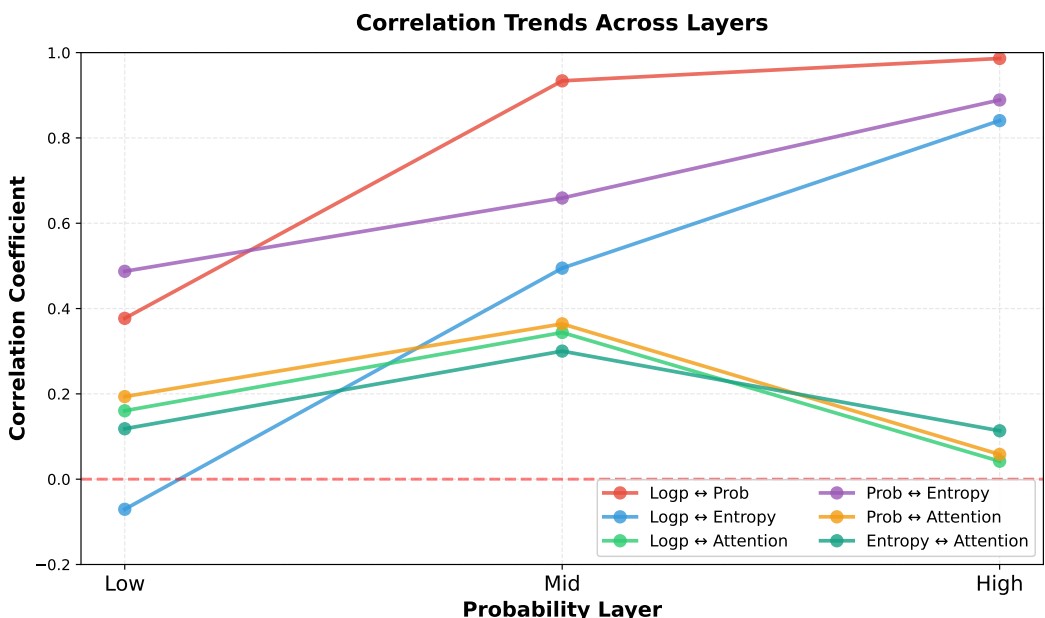

Figure 24: Correlation comparison across layers between different image token selection strategies over the wemath benchmark.

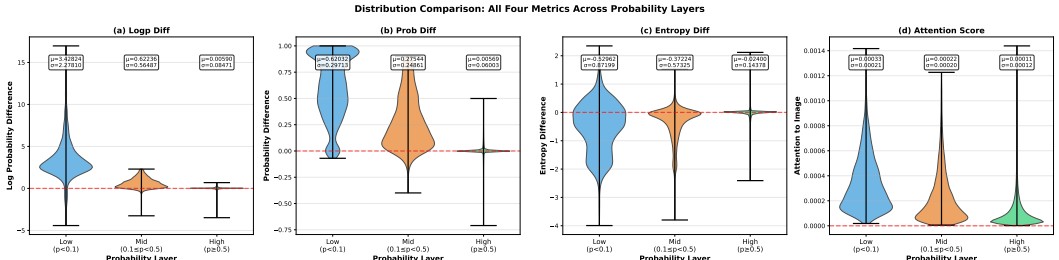

Figure 25: violin plot between different image token selection strategies over the validation set.

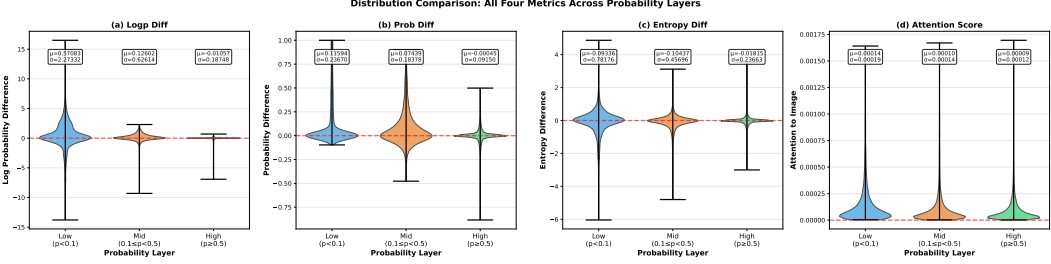

Figure 26: violin plot between different image token selection strategies over the hallubench benchmark.

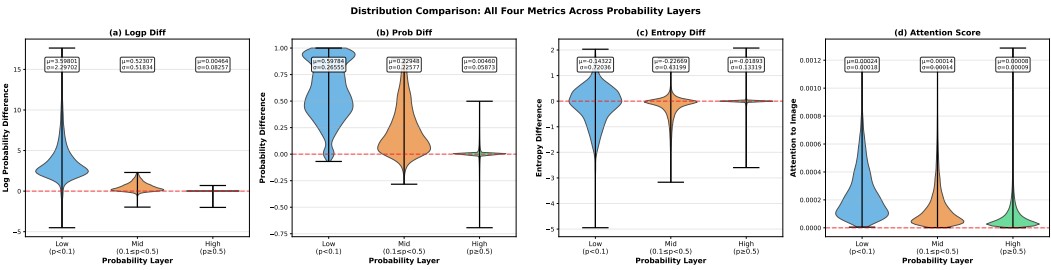

Figure 27: violin plot between different image token selection strategies over the wemath benchmark.

Figure 28: Distribution of selected tokens over the sampled rollouts, where we employ the Qwen-VL-2.5 7B model with a batch of 512 samples, each sample generates 16 rollout responses.

