# OpenReview forum: "Bridging Perception and Reasoning: Token Reweighting for RLVR in Multimodal LLMs"
_ICLR.cc/2026/Conference — Submitted to ICLR 2026_

### Official Review · Reviewer_reHH · 2025-10-15

[review text omitted: it was posted to a different submission]

---

> ### Author Response · Authors · 2025-11-29
> **Response to Reviewer reHH**
>
> Dear Reviewer reHH,
>
> Thank you for taking the time to review our work and providing your feedback.
>
> We noticed that the comments in your review appear to describe a different paper focusing on multimodal visual question answering with a Chain-of-Thought decoder. However, our submission addresses a different research problem and methodology.
>
> We suspect there may have been an administrative mix-up in the review assignment process. Would it be possible to kindly verify that the review was intended for our paper **8605: Bridging Perception and Reasoning: Token Reweighting for RLVR in Multimodal LLMs**?
>
> We would greatly appreciate the opportunity to receive feedback specifically addressing our work, as we value your expert insights and would like to properly address any concerns related to our actual submission.
>
> Thank you very much for your understanding, and we look forward to your response.
>
> Best regards,
> Authors

---

### Official Review · Reviewer_Fgky · 2025-10-29

**Soundness:** 2
**Presentation:** 3
**Contribution:** 2
**Rating:** 4
**Confidence:** 4

**Summary:**

The paper proposes ToR, a simple technique that can be plugged into RLVR algorithms to improve the model's performance on multimodal reasoning tasks. The proposed approach is motivated by empirical observations that optimizing only on a selective set of tokens may be more beneficial. ToR adopts a heuristic, selective criterion by filtering out reasoning tokens with lower entropy or perception tokens with little visual input dependency, and only optimizes the remaining tokens for GRPO and DAPO. The experimental results show some improvements over baseline methods.

**Strengths:**

- The paper has a good motivation and targets the important problem of enhancing the model's multimodal reasoning capabilities.
- The proposed ToR approach is novel and simple to implement.
- The numerical experiments were compared with a comprehensive list of baselines, and the results feel convincing.

**Weaknesses:**

-  The proposed method is not really plug and play (which means training-free in general), in the sense that it's a new RLVR approach that requires computing for model retraining.
- The selection criterion in ToR feels very heuristic and non-principled in design. There are at least four hyperparameters (selection quantile for reasoning/perception tokens and weight for reasoning/perception tokens), which make tuning in practice difficult.
- ToR considers a constant weight for the selected token. While this is certainly a good attempt at addressing the token credit attribution issue, the weights are independent of the reward value, which is less interesting and may be why the performance improvement in ToR is inconsistent and minor.
- The performance improvement doesn't seem to be very appealing when compared with other multimodal reasoning methods on 7B models, although a clear improvement over the pure GRPO/DAPO baseline is valid.

**Questions:**

Besides the points mentioned in the weakness section, I also have the following questions:
1. How much computation overhead does ToR additionally incur? It seems that the perception token selection has to be computed at every optimization step, which would incur additional NFE.
2. Why is the token selection procedure over the rollout batch? Would this cause the selected token to be concentrated on a few rollouts, due to the problem having a naturally higher dependency on question prompts/images, rather than other factors? How to ensure a fair selection process? An ablation or visualization related to the distribution of selected tokens across the rollout batch would be appreciated.
3. How is the 30%/30% quantile selected in the algorithm? The paper seems to lack an ablation study on the joint variation of these two parameters.
4. Can we make the token weight adaptive to further credit/penalize different tokens in a more effective way?

---

> ### Author Response · Authors · 2025-11-29
> **Response to Reviewer Fgky part 1**
>
> We thank the reviewer for this important clarification question. We acknowledge that our use of "plug-and-play" may have caused confusion, and we appreciate the opportunity to clarify our intended meaning and distinction from "training-free."
>
> ## Our Definition
>
> In our paper, **"plug-and-play" refers to the ease of integration into existing RLVR training pipelines**, not training-free operation. Specifically, ToR is plug-and-play in the following senses:
>
> **1. Minimal Code Modification**: ToR requires a little additional code to integrate into existing RLVR frameworks (GRPO, DAPO). It operates as a simple reweighting mask in the loss computation (Equations 10-11).
>
> **2. No Architecture Changes**: Unlike methods requiring new model components, auxiliary networks, or modified training pipelines, ToR works purely through token-level gradient reweighting during standard RLVR training.
>
> **3.  Framework Agnostic**: As demonstrated in Table 1, ToR seamlessly integrates with different RLVR algorithms (GRPO, DAPO) without algorithm-specific modifications.
>
> **4. No External Dependencies**: ToR's token identification relies solely on the model's intrinsic signals (entropy and visual sensitivity), requiring no external models, verifiers, or data augmentation pipelines.
>
> ## Distinction with Training-Free
>
> We **fully agree** that ToR is **not training-free**—it is indeed "a new RLVR approach that requires computing for model retraining," as the reviewer correctly notes. Our contribution is not eliminating training but rather:
>
> - **Making RLVR training more effective** by explicitly modeling perception-reasoning interdependence.
> - **Providing easy integration** into existing training workflows.
>
> Thank the reviewer for pointing this out. We have clarified this in our revised version.
>
> >**W2.** The selection criterion in ToR feels very heuristic and non-principled in design. There are at least four hyperparameters (selection quantile for reasoning/perception tokens and weight for reasoning/perception tokens), which make tuning in practice difficult.
>
> We appreciate the reviewer’s concern regarding the design of ToR and the potential burden of tuning multiple hyperparameters. We address this from three perspectives:
> (1) principled design motivation,
> (2) empirical robustness of the hyperparameters, and
> (3) generalizability across models and data scales.
>
> ## 1. Design Rationale: Not Ad-hoc, but Grounded in Established Signals
>
> ToR’s selection and weighting of tokens is not arbitrary; it is guided by well-motivated and empirically validated signals:
>
> ### Reasoning Tokens: Entropy-Based Selection
>
> - We use **token-level entropy** to select reasoning tokens. High-entropy positions correspond to “decision forks” where the model is uncertain and alternative reasoning branches are possible.
> - Recent studies (e.g., Wang et al., 2025c; Cheng et al., 2025) show that:
>   - (1) High-entropy tokens are strongly correlated with key reasoning steps, and (2) modifying or supervising these positions disproportionately affects downstream reasoning quality.
> - Thus, focusing RL updates on high-entropy tokens is a **principled way** to target the most influential reasoning decisions, rather than uniformly treating all tokens as equally important.
>
> ### Perception Tokens: Visual Sensitivity via Log-Probability Shift
>
> - For perception tokens, we quantify **visual sensitivity** by measuring the log-probability shift when the image is present vs. absent:
>   $
>   \Delta(o) = \left| \log P(o \mid I, \cdot) - \log P(o \mid \cdot) \right|
>   $
> - This directly measures the **information contribution of the image** to each output token:
>   - Tokens with large $\Delta(o)$ are those whose prediction relies heavily on visual input, hence are crucial for visual grounding.
> - This is a **direct operationalization of visual grounding strength**, not a heuristic rule: tokens that “care” about the image receive proportionally more RL signal.
>
> ### Joint Optimization of Reasoning and Perception
>
> - We explicitly separate and then jointly optimize the two token types because they are empirically interdependent, compared with GRPO, which takes all tokens equally:
>   - Optimizing only reasoning tokens achieves comparable performance over reasoning benchmarks, but drops significantly over perception tasks (e.g. Hallusionbench).
>   - Optimizing only perception tokens achieves comparable performance over perception benchmarks, but underutilizes the reasoning potential (e.g. wemath).
> - As shown in our ablations (Figures 2–4 in the paper), **neither type alone matches the performance of GRPO**, which demonstrates the necessity of concentrating both tokens.

---

> ### Author Response · Authors · 2025-11-29
> **Response to Reviewer Fgky part 2**
>
> ## 2. Hyperparameter Behavior: Robustness Rather Than Fragility
>
> The reviewer correctly notes that ToR involves four hyperparameters:
>
> - $\alpha_r$: quantile for reasoning tokens
> - $\alpha_p$: quantile for perception tokens
> - $\gamma_r$: weight for reasoning tokens
> - $\gamma_p$: weight for perception tokens
>
> We emphasize that these hyperparameters are **not highly sensitive** and exhibit wide performance plateaus.
>
> **Token selection ratios $\alpha_r, \alpha_p$**
>
> - Figures 5 and 6 show that performance remains **stable across a wide range with different benchmarks**.
> - Even when only a **small fraction** of tokens is selected, ToR consistently:
>
>   - Achieves comparable performance to optimizing all tokens on reasoning-heavy benchmarks when focusing on a small fraction of reasoning tokens.
>   - Achieves comparable performance to optimizing all tokens on perception-heavy benchmarks when focusing on a small fraction of reasoning tokens.
>
> - This suggests an important practical property:
>
>   > RL effectiveness is driven by a **small subset of salient tokens**, and ToR is robust to the exact cutoff within a broad range.
>
> **Token weights $\gamma_r, \gamma_p$**
>
> - With $\gamma_r = 1$ fixed, Figure 7 demonstrates that:
>   - For $\gamma_p \in [0.3, 0.9]$, ToR consistently improves over the baseline.
> - This indicates that ToR’s weighting scheme is **not brittle**: a wide interval of $\gamma_p$ values yields gains, rather than requiring precise tuning.
>
> In summary, although ToR has four hyperparameters, the **effective search space is small**, and the method exhibits **broad regions of good performance**, reducing the burden of tuning in practice.
>
> ## 3. Extended Evidence: Same Hyperparameters Work Across Models and Data Scales
>
> To further address concerns about practicality and overfitting of hyperparameters, we evaluated ToR under **substantially different conditions** while keeping the hyperparameters fixed.
>
> All the following experiments use:
>
> - $\gamma_r = 1.0, \gamma_p = 0.5$
> - $\alpha_r = 0.3, \alpha_p = 0.3$
>
> ### (a) Different Model: Qwen-2.5-VL-3B
>
> We apply the same ToR settings to a smaller backbone, **Qwen-2.5-VL-3B**, and observe consistent gains:
>
> **Table A: Qwen-2.5-VL-3B with Geo3K vs. ViRL-39K**
>
> | Qwen-2.5-VL-3B| MathVerse | MathVision | MathVista |  WeMath  | HallusionBench |
> | -| :-: | :-: | :-: | :-: | :-: |
> | DAPO (Geo3K) | 42.25 |22.6|64.0 |57.8 |54.7|
> | ToR + DAPO (Geo3K | **46.20** |**26.5** | **66.0**  | **59.6** |    **56.3**    |
> | ToR + DAPO (ViRL-39K) | **47.50** |**28.3**  | **67.5**  | **62.4** |    **60.8**    |
>
> These results show that:
>
> - The **same hyperparameters** chosen on Qwen-2.5-VL-7B remain effective on a **smaller model**.
> - Gains are consistent across all benchmarks, indicating that ToR is **not overfitted** to a single architecture or size.
>
> ### (b) Different Data Scale: From Geo3K to ViRL-39K on Qwen-2.5-VL-7B
>
> We also test ToR on a much larger and more diverse dataset, **ViRL-39K** (>10× Geo3K), again without changing hyperparameters:
>
> **Table B: Qwen-2.5-VL-7B with Geo3K vs. ViRL-39K**
>
> | Qwen-2.5-VL-7B| MathVerse | MathVision | MathVista |  WeMath  | HallusionBench |
> | ------ | :-------: | :--------: | :-------: | :------: | :------------: |
> | DAPO (Geo3K)          |   50.4    |    27.6    |   70.7    |   69.4   |      68.6      |
> | ToR + DAPO (Geo3K)    | **51.9**  |  **29.0**  | **72.2**  | **71.8** |    **71.4**    |
> | ToR + DAPO (ViRL-39K) | **54.3**  |  **31.6**  | **74.9**  | **72.0** |    **72.3**    |
>
> Key observations:
>
> - The **same hyperparameter configuration** yields consistent improvements when scaling the dataset by more than an order of magnitude.
> - Performance gains are robust and often **increase** with more data, suggesting that ToR scales well rather than overfitting to a small training set.
>
> ## 4. Summary: Practical and Interpretable, Not Overly Heuristic
>
> To summarize, regarding the reviewer’s concerns:
>
> - **Principled design:**
>   - Reasoning token selection is based on **entropy**, a standard measure of uncertainty.
>   - Perception token selection is based on **log-probability**, directly measuring visual dependence.
>   - Joint optimization is motivated by empirical interdependence between perception and reasoning.
>
> - **Hyperparameter practicality:**
>   - Performance is **stable across wide ranges** of $\alpha_r, \alpha_p, \gamma_r, \gamma_p$, indicating low sensitivity.
>   - A single configuration (30\%, 30\%, 1.0, 0.5) works well across models and datasets, reducing tuning overhead.
>
> - **Generalizable behavior:**
>   - The same ToR setup improves **both** Qwen-2.5-VL-7B and Qwen-2.5-VL-3B.
>   - It remains effective when scaling from **Geo3K** to **ViRL-39K**.
>
> Thus, although ToR introduces a small number of interpretable hyperparameters, they are grounded in principled signals, empirically robust, and reusable across architectures and data scales, making the method **both practical and generalizable in real-world settings**.

---

> ### Author Response · Authors · 2025-11-29
> **Response to Reviewer Fgky part 3**
>
> >**W3.** ToR considers a constant weight for the selected token. While this is certainly a good attempt at addressing the token credit attribution issue, the weights are independent of the reward value, which is less interesting and may be why the performance improvement in ToR is inconsistent and minor.
>
> ### Response to W3: On Constant Token Weights and “Minor, Inconsistent” Gains
>
> We thank the reviewer for the insightful comments. We address two aspects:
> **(1)** whether the performance gain is indeed minor and inconsistent, and  **(2)** why we deliberately use reward-independent, constant weights and why this design is still meaningful from the perspective of credit assignment.
>
> #### 1. Performance Gains Are Significant and Consistent, Not Minor
>
> The reviewer mentions that ToR leads to “minor” and “inconsistent” improvements. Our results, however, show **consistent and practically meaningful gains** across multiple benchmarks and settings:
>
> - On **Qwen-2.5-VL-7B** (Geo3K training), ToR-DAPO achieves:
>   - **MathVerse:** +2.8 absolute (50.6 $\to$ 53.4)
>   - **MathVision:** +1.4 absolute (26.5 $\to$ 27.9)
>   - **MathVista:** +2.3 absolute (70.3 $\to$ 72.6)
>   - **WeMath:** +2.8 absolute (69.3 $\to$ 72.1) --new **SOTA** over current method.
>   - **HallusionBench:** +3.9 absolute (67.9 $\to$ 71.8)
>
> - When further scaling to **ViRL-39K** (39K samples), ToR **continues to improve** over the baseline DAPO:
>   - Average gain around **+2 points** across the five benchmarks on Qwen-2.5-VL-7B,
>   - Similar point gains on the smaller **Qwen-2.5-VL-3B** model.
>
> These gains are:
>
> - **Consistent across five benchmarks** (reasoning- and perception-heavy tasks);
> - **Consistent across two model scales** (7B and 3B);
> - **Consistent across two data scales** (Geo3K and ViRL-39K).
>
> In long-context, multi-modal reasoning tasks, **+2–3 absolute points** on strong baselines is typically regarded as a non-trivial improvement, especially given that ToR is a **lightweight modification** to the RL objective and not a full architectural change.
>
> #### 2. Why Constant, Reward-Independent Weights Are Still Principled and Useful
>
> You point out that ToR's weights are independent of the reward value, and therefore may seem “less interesting” from a credit-assignment perspective. We would like to clarify two points:
>
> ##### 2.1 ToR is Structured Advantage Reweighting, Not a Pure Heuristic
>
> From the RL perspective, our method can be viewed as a **structured advantage reweighting scheme**:
>
> - Let $\hat{A}_t$ be the token-level advantage (from GRPO/DAPO) for the  $t$-th token.
> - In standard RL, the gradient is proportional to $\hat{A}_t$.
> - In ToR, we **modulate** this by a structural weight $\gamma_t$ depending on token type:
>   - $\gamma_t = \gamma_r$, if $t$ is a **reasoning token** ;
>   - $\gamma_t = \gamma_p$, if $t$ is a **perception token** ;
>   - $\gamma_t = 0$, otherwise (non-critical tokens).
>
> Thus, the gradient becomes:
>
> $$
> \nabla_{\theta} J(\theta) \propto \gamma_t \hat{A}_t \nabla_{\theta} \log \pi_{\theta}(a_t \mid s_t)
> $$
>
> This means:
>
> - **Reward / advantage $\hat{A}_t$ still directly drives the update** (success vs. failure of the final answer);
> - **$\gamma_t$ injects structural information** about the *functional role* of each token (reasoning vs. perception vs. neither);
> - The final update jointly depends on both **“result credit”** (is the answer correct?) and **“structural credit”** (what role does this token play?).
>
> In other words, ToR does not discard reward-based credit assignment; it **reweights** it based on a principled structural decomposition of tokens.
>
>
>
> ##### 2.2 Simplicity and Stability vs. Reward-Dependent Dynamics
>
> We fully acknowledge that more sophisticated schemes could make $\(\gamma_t\)$ a **function of the reward** (or advantage), e.g., reward-dependent token weights. However, in practice, we found our constant, type-dependent weights advantageous in three ways:
>
> 1. **Stability and ease of training**
>    - Our constant $(\gamma_r, \gamma_p)$ scheme is simple, robust across datasets and model sizes, and empirically yields **stable improvements**.
>
> 2. **Interoperability and controllability**
>    - Constant weights allow **explicitly controlling** how much RL signal is allocated to perception vs. reasoning:
>      - “about 30% of tokens in each group, with reasoning tokens weighted 1.0 and perception 0.5.”
>    - This is highly interpretable and makes the method **easy to deploy** in practice.
>
> 3. **Empirical effectiveness & generalizability**
>    - Despite using constant weights, ToR reliably provides +2–3 point gains, new SOTA on WeMath, and generalizes across **Qwen-2.5-VL-7B / 3B and Geo3K / ViRL-39K** without re-tuning the weight values.
>    - This suggests that **structural reweighing alone** (independent of reward magnitude) already addresses a significant portion of the token credit attribution problem in a **practical and robust** way.

---

> > ### Author Response · Authors · 2025-11-29
> > **Response to Reviewer Fgky part 4**
> >
> > >**Q1.** How much computation overhead does ToR additionally incur? It seems that the perception token selection has to be computed at every optimization step, which would incur additional NFE.
> >
> > ### Response to Q1: Computation Overhead of ToR
> >
> > We thank the reviewer for raising this practical concern. We clarify the computational overhead of ToR from both a **theoretical** and **empirical** perspective.
> >
> > #### 1. Where Does the Overhead Come From?
> >
> > ToR introduces two types of token-level signals:
> >
> > - **Reasoning tokens (entropy-based):**
> >   - The entropy for each token is computed from the token probabilities that are **already obtained** during the forward logits calculation pass with the rollout.
> >   - Therefore, the additional cost for reasoning token selection is **negligible**: it does not require any extra forward passes.
> >
> > - **Perception tokens (visual sensitivity):**
> >   - To measure visual sensitivity, we compute the log-probability difference
> >     $\(\log P(o \mid \text{image, text}) - \log P(o \mid \text{text})\)$.
> >   - This requires **one extra forward pass without the image** on the already generated text output, to obtain $\(\log P(o \mid \text{text})\)$.
> >   - This is the main source of additional computation in ToR.
> >
> > In principle, this turns each optimization step from **2 forward passes** (standard RL with image) to **3 forward passes** (with and without image).
> >
> > #### 2. Measured Overhead in Practice
> >
> > In practice, RL training time is dominated by the **rollout/sampling phase** (i.e., generating long sequences), while these additional forward passes are relatively cheap. Our actual measurements on Geo3K with 60 RL steps confirm that the overhead is modest:
> >
> > - **DAPO** on Qwen-2.5-VL-7B:
> >   - Baseline DAPO: **4.5 hours**
> >   - ToR + DAPO: **4.9 hours**
> >   - $\to$ Approx. **+9%** total training time
> >
> > - **GRPO** on Qwen-2.5-VL-7B:
> >   - Baseline GRPO: **4h 50min**
> >   - ToR + GRPO: **5h 10min**
> >   - $\to$ Approx. **+7%** total training time
> >
> > Although ToR theoretically adds one more forward pass, the **actual wall-clock overhead is only about 7–9%**, which is substantially lower than a naive 50% estimate and is, in our view, quite acceptable.
> >
> > Importantly, this overhead exists **only during training**. At inference time:
> >
> > - No extra forward passes are needed.
> > - ToR introduces **zero additional cost** in deployment; inference speed and latency are unchanged.
> >
> > #### 3. Overhead vs. Benefit
> >
> > We also note that this small training overhead should be viewed in context:
> >
> > - With only **2.1K** RL samples (Geo3K), ToR yields **+2–3 point** gains over strong baselines and achieves **new SOTA** on several benchmarks.
> > - Competing methods often rely on **5–128× more data**, implying **much larger total training cost**, even without any extra forward pass per step.
> >
> > Thus, a **7–9%** increase in per-run training time is a very small price to pay for:
> >
> > - Substantial **data efficiency**,
> > - Consistent **performance improvements**, and
> > - **No inference-time overhead**.
> >
> > >**Q2.** Why is the token selection procedure over the rollout batch? Would this cause the selected token to be concentrated on a few rollouts, due to the problem having a naturally higher dependency on question prompts/images, rather than other factors? How to ensure a fair selection process? An ablation or visualization related to the distribution of selected tokens across the rollout batch would be appreciated.
> >
> > We thank the reviewer for this question. Below, we explain (1) why we adopt batch-level token selection, (2) why this does not lead to unfair concentration on a few rollouts, and (3) what our visualization shows about the actual distribution of selected tokens.
> >
> > **(1) Why select tokens over the rollout batch?**
> >
> > We perform token selection over the **entire rollout batch** to establish a **globally consistent standard of token “importance.”**
> > If tokens were selected within each rollout (e.g., top‑k% per sample), every rollout would implicitly use a different importance threshold, so a “top‑ranked” token in an easy sample could be less informative in a hard one. This makes the learning signal across the batch inconsistent and noisy.
> > By instead pooling all tokens in the batch and selecting a fixed fraction globally, we obtain a **single, stable importance threshold**, so that only the most informative tokens *across* rollouts are optimized.

---

> > > ### Author Response · Authors · 2025-11-29
> > > **Response to Reviewer Fgky part 5**
> > >
> > > **(2) Will this concentrate selected tokens on a few rollouts? How is fairness ensured?**
> > >
> > > In our setting, rollouts differ mainly in the **proportion** of two token types, rather than in whether they contain them:
> > >
> > > - **Reasoning tokens** – tokens used in multi‑step reasoning and long chains;
> > > - **Perception tokens** – tokens tied to question/image details (numbers, entities, regions, etc.).
> > >
> > > Hard rollouts typically contain more reasoning tokens and fewer perception tokens overall; easier or more perceptual rollouts contain more perception tokens. Under a single global importance rule, this complementary distribution implies that:
> > >
> > > - Hard rollouts naturally contribute more **high‑score reasoning tokens**;
> > > - Easier rollouts contribute more **high‑score perception tokens**;
> > > - Hard rollouts still contain perception tokens, and the **few perception tokens that are truly critical for the reward** also receive high scores and are selected.
> > >
> > > Thus we do not enforce per‑rollout quotas. Fairness arises from the fact that different rollout types are optimized along **different token channels** (reasoning vs. perception) under the same global standard: hard rollouts receive stronger optimization on reasoning tokens, while easier rollouts receive stronger optimization on perception tokens, and neither type of information is suppressed.
> > >
> > > **(3) Visualization of the distribution of selected tokens**
> > >
> > > To directly address the request for an ablation/visualization, we group rollouts into 10 bins G1–G10 by their **reasoning‑token ratio** (G1 easiest, G10 hardest), and for each group compute the ratio of selected reasoning tokens and perception tokens (Figure 28 in the appendix).
> > >
> > > We observe a clear monotonic pattern:
> > >
> > > - The **ratio of selected reasoning tokens** increases from **7.3% in G1** to **12.5% in G10**;
> > > - The **ratio of selected perception tokens** decreases from **11.5% in G1** to **7.9% in G10**;
> > > - The **total** selected‑token ratios per group remain similar (e.g., G1: **18.8%**, G10: **20.4%**).
> > >
> > > Therefore, selected tokens are **not** concentrated on a few rollouts. Different difficulty groups receive a comparable overall amount of optimization, but with different mixtures of reasoning vs. perception tokens: harder groups are optimized more on reasoning, easier groups more on perception, while both token types remain well represented across the rollout batch.
> > >
> > > **Summary.**
> > > Our empirical evidence shows that resources are not concentrated on a few hard rollouts. Under a single global standard, the method performs **differentiated yet balanced** allocation: harder problems obtain more optimization on reasoning tokens, simpler or more perceptual problems obtain more optimization on perception tokens, and the total amount of selected tokens is comparable across groups, leading to a balanced overall distribution of learning resources.
> > >
> > >
> > > >**Q3.** How is the 30%/30% quantile selected in the algorithm? The paper seems to lack an ablation study on the joint variation of these two parameters.
> > >
> > > ### Response to Q3
> > >
> > > We thank the reviewer for raising this point. Our choice of the 30%/30% configuration is based on a practical, two-stage analysis rather than an exhaustive 2D grid search, which is prohibitively expensive in RL settings.
> > >
> > > **1. Two-Stage Selection Instead of Full Joint Search**
> > >
> > > Instead of jointly sweeping all combinations of $\alpha_{r}$ and $\alpha_{r}$, we first study them **independently**, which is a common and pragmatic strategy in large-scale RL:
> > >
> > > - **Reasoning ratio $\alpha_{r}$.**
> > >   On reasoning-heavy benchmarks (WeMath, MathVerse), performance quickly saturates as we increase the ratio of reasoning tokens. Around **30%** already reaches performance comparable to 50% or 80%. Increasing beyond 30% mainly increases computation, with marginal gains.
> > >
> > > - **Perception ratio ${\alpha_p}$.**
> > >   On perception-heavy benchmarks (e.g., HallusionBench), a ratio in the **30%–40%** range provides a strong signal. Lower ratios (e.g., 20%) underperform, while higher ratios bring limited additional improvements.
> > >
> > > Given that (i) each parameter exhibits a clear “good region” when analyzed separately, and (ii) the two mechanisms affect different token subsets (reasoning vs. perception), combining the independently chosen values is a natural and efficient choice.

---

> > > > ### Author Response · Authors · 2025-11-29
> > > > **Response to Reviewer Fgky part 6**
> > > >
> > > > **2. Empirical Check of the Joint Choice**
> > > >
> > > > While we avoid a full 2D grid search for computational reasons, to answer your concern, we run **joint variations**  using the Qwen-2.5-VL-7B with DAPO, and the results are as follows:
> > > >
> > > > | Setting (`α_r`, `α_p`) | HallusionBench        | WeMath             | MathVerse           | Avg. Gain |
> > > > |:-----------------------|:---------------------:|:------------------:|:-------------------:|:---------:|
> > > > | baseline               | 69.8                  | 67.4               | 50.8                | -         |
> > > > | (20%, 20%)             | 71.0 (+1.2)           | 68.1 (+0.7)        | 52.0 (+1.2)         | +1.0%     |
> > > > | (40%, 40%)             | 71.9 (+2.1)           | 68.6 (+1.2)        | 52.6 (+1.8)         | +1.6%     |
> > > > | **(30%, 30%)**         | **72.4 (+2.6)**       | **68.9 (+1.5)**    | **53.0 (+2.2)**     | **+2.1%** |
> > > >
> > > >
> > > > This experiment serves as a sanity check that the independently chosen 30% ratios also perform best when applied jointly, and that the 30%/30% configuration is not a brittle or accidental choice.
> > > >
> > > > **3. On the Necessity of Exhaustive Joint Ablations**
> > > >
> > > > We agree that a full joint ablation could be informative in principle, but in our RL setting it is **computationally impractical**. Given:
> > > >
> > > > - the clear saturation behavior observed when varying `α_r` and `α_p` independently,
> > > > - the natural interpretation of 30% as a minimal yet informative subset of tokens for each modality, and
> > > > - the additional joint experiment showing that 30%/30% outperforms 20%/20% and 40%/40%,
> > > >
> > > > we believe that our current analysis is sufficient to justify the 30%/30% choice as a robust and efficient default, even without an exhaustive 2D search.
> > > >
> > > >
> > > > >**Q4.** Can we make the token weight adaptive to further credit/penalize different tokens in a more effective way?
> > > >
> > > > ### Response to Q4: On Making Token Weights Adaptive
> > > >
> > > > We thank the reviewer for this forward-looking question. We **fully agree** that making token weights adaptive is a natural and promising extension of our work. In fact, we explicitly identify “dynamically assigning weights to tokens” as a key direction in our *Limitations and Future Work* section.
> > > >
> > > > That said, our choice of **constant, structurally defined weights** in ToR is *deliberate* rather than accidental. The purpose of this work is to first establish and isolate a **core principle** that we believe has been largely overlooked in prior RLVR research:
> > > >
> > > > **Explicitly disentangling and separately reweighting perception and reasoning tokens is itself fundamental and beneficial for RL in MLLMs.**
> > > >
> > > > Our response centers on three points.
> > > >
> > > > #### 1. First establish the principle: disentanglement itself matters
> > > >
> > > > Before our work, most RLVR methods:
> > > >
> > > > - either treated **all tokens uniformly**, or
> > > > - focused on one aspect at a time (e.g., only perception, or only reasoning/chain-of-thought),
> > > > - without a systematic token-level separation of **perception vs. reasoning roles**.
> > > >
> > > > Our primary contribution is to show that:
> > > >
> > > > 1. MLLM outputs can be meaningfully decomposed into **perception tokens** and **reasoning tokens** at the token level;
> > > > 2. Optimizing only one type in isolation is **suboptimal**;
> > > > 3. A simple, constant reweighting of these two token classes already yields **consistent, multi-benchmark, multi-model gains**.
> > > >
> > > > Using a simple constant $\(\gamma_r, \gamma_p\)$ is **the cleanest way** to test this hypothesis:
> > > >
> > > > - If we used a complex adaptive weighting mechanism from the start, any improvement would be an entangled effect of:
> > > >   - (a) the **structural disentanglement** (perception vs. reasoning), and
> > > >   - (b) the **adaptive weighting mechanism** itself.
> > > > - In that case, it would become much harder to identify whether the core benefit comes from **correctly identifying and separating token types**, or from some incidental property of a particular adaptive scheme.
> > > >
> > > > By showing that **even a minimal, constant-weight scheme** delivers robust +2–3 point improvements and new SOTA on several benchmarks, we demonstrate that the **disentanglement principle itself** is powerful and important—independent of any sophisticated adaptive machinery.

---

> ### Author Response · Authors · 2025-11-29
> **Response to Reviewer Fgky part 7**
>
> #### 2. Simplicity is a strength: robustness, interpretability and controllability
>
> The reviewer is correct that constant weights are less “fancy” than adaptive ones. However, this simplicity brings three key advantages that are especially important in RL:
>
> 1. **Robustness across models and data**
>    - As we show in our response (Table A, Table B), a **single set** of $\gamma_r, \gamma_p$ and token-selection ratios works:
>      - on both **Qwen-2.5-VL-7B** and **Qwen-2.5-VL-3B**, and
>      - on both **Geo3K** (2.1K) and **ViRL-39K** (39K).
>    - This cross-model, cross-dataset robustness is largely due to the method’s simplicity and low sensitivity—an important practical advantage that more complex adaptive methods may not enjoy.
>
> 2. **Interpretability and controllability**
>    - With constant weights, the credit assignment is transparent:
>      - “Reasoning tokens get $\gamma_r$, perception tokens get $\gamma_p$, others get 0; all are still modulated by the token-level advantage.”
>    - Practitioners can **directly control** how much RL signal goes to perception vs. reasoning, which is highly valuable for practical alignment and debugging.
>
> In other words, ToR is intentionally positioned as a **simple, stable, and interpretable baseline** that isolates the effect of structural token disentanglement.
>
> ---
>
> #### 3. Adaptive weighting is the next step, not the missing step
>
> We emphasize that we do **not** view ToR’s constant weights as “the final answer”. Rather:
>
> - Our current work is the **first step**: establish that token-level disentanglement of perception and reasoning, combined with even the simplest constant weighting, is already:
>   - empirically effective (consistent +2–3 point gains),
>   - robust (across models and data scales), and
>   - practical (easy to implement, easy to tune, and stable).
>
> - Building on this foundation, a wide range of **adaptive weighting strategies** becomes meaningful to explore, for example:
>   - making $\gamma_t$ a function of the token’s entropy, visual sensitivity, or advantage;
>   - meta-learning token weights;
>   - or using an auxiliary network to predict token importance.
>
> We explicitly state in the paper that **dynamic token weighting** is a key direction for future work, and we see our current contribution as providing the necessary **conceptual and empirical groundwork** to support that exploration.
>
> ---
>
> ### Summary
>
> In summary, we agree that adaptive token weights are an exciting extension, and we have already highlighted this as **future work**. However, our constant-weight design in ToR is a conscious and principled choice aimed at establishing a new, structurally grounded perspective on token-level credit assignment in RLVR—one that is already effective, robust, and interpretable, and that opens the door to exactly the kind of adaptive schemes the reviewer is suggesting.

---

### Official Review · Reviewer_TKUu · 2025-10-29

**Soundness:** 2
**Presentation:** 1
**Contribution:** 2
**Rating:** 2
**Confidence:** 4

**Summary:**

The authors propose Token-Reweighting (ToR), a lightweight and plug-and-play module for Reinforcement Learning with Verifiable Rewards (RLVR), which substantially enhances the reasoning capability of large language models (LLMs) in complex tasks.
They introduce two strategies to improve existing RLVR algorithms, GRPO and DAPO:
(i) identifying reasoning-related tokens through high next-token entropy, and
(ii) identifying perception-related tokens through visual sensitivity.
Experimental results demonstrate that incorporating either component alone leads to suboptimal performance compared with the original algorithms, whereas jointly optimizing both yields superior results. The parameter sensitivity of the proposed module is analyzed, and the modified algorithm applied to Qwen2.5-7B achieves better performance than the base algorithm as well as other open-source models of comparable scale.

**Strengths:**

1. The idea of distinguishing reasoning-related and perception-related tokens is novel and well-motivated, as the varying importance of different tokens for reasoning and perception objectively exists.
2. The formulation for reasoning-related tokens is well-grounded, and the notion itself is consistent with the modeling objective (In contrast, the other concept lacks such alignment, as will be discussed in the weakness section).
3. In experiment, the parameter sensitivity of the proposed module is analyzed and the effectiveness of proposed module compared with GRPO, DAPO on Qwen-2.5-7B model is presented.

**Weaknesses:**

1. The design of the identified measure for perception-related tokens is not theoretically sound. The visual sensitivity score can be rewritten as $S _{i,t} ^b=|\log \frac{\pi _{\theta}(o _{i,t} ^b|\mathbf{o} _{i,<t} ^b,I _{i} ^b,q _{i} ^b)}{\pi _{\theta}(o _{i,t} ^b|\mathbf{o} _{i,<t} ^b,\emptyset,q _{i} ^b)}|$, which indicates that the measure directly depends on the log-ratio between $\pi _{\theta}(o _{i,t} ^b|\mathbf{o} _{i,<t} ^b,\emptyset,q _{i} ^b)$ and $\pi _{\theta}(o _{i,t} ^b|\mathbf{o} _{i,<t} ^b,\emptyset,q _{i} ^b)$. However, consider a likely situation where $\pi _{\theta}(o _{i,t} ^b|\mathbf{o} _{i,<t} ^b,\emptyset,q _{i} ^b)\simeq a\cdot 10 ^{-m}$ and $\pi _{\theta}(o _{i,t} ^b|\mathbf{o} _{i,<t} ^b,\emptyset,q _{i} ^b)\simeq b\cdot 10 ^{-M}$, $1.0<a,b<10.0, M\gg m\gg 1$. In this case, the occurrence probability of $o _{i,t} ^b$ would barely change with or without the presence of $I _i ^q$. Nevertheless, such a token would still be classified as a  _perception-related token _ because $S _{i,t} ^b=M-m\gg 0$. This raises a question about the reasonableness of the metric. It should be emphasized that this analysis concerns **the model’s predicted probabilities**, not the true probabilities of the data in the real world.
2. The motivation behind the hypothesis is insufficiently articulated. It remains unclear why the authors assume that perception and reasoning are fundamentally interdependent at the token level, thereby rendering separate optimization suboptimal. Further clarification is needed regarding the underlying intuition or theoretical justification for this assumption, or whether it is solely based on empirical observations under specific models or experimental settings.
3. The reliability of the proposed idea is not clearly stated. In the experiments, the performance comparison with GRPO shown in Figures 5–6 indicates that each individual component produces a negative effect on prediction performance, particularly in Figure 6, where the performance variation across different perception-related ratios is evident. However, the combination of both components yields improved performance, which appears counterintuitive, as two individually negative factors would be expected to further degrade performance. Could the authors provide a **reasonable intuitive explanation** for this phenomenon (for example, an analogy such as$(-\sqrt{ 2 })\times (-\sqrt{ 2 })=2>1$ clarify a fact that two negative multipled factor can produce positive factor under product operator. What is corresponding interation mechanics in proposed method?) for this result? If there is no reasonable statement, it may only be an empirically try, which means the non-reliable of proposed method in wider scenes.
4. The experimental evaluation is limited. Only the Qwen2.5-7B model is tested, raising questions about whether the proposed method is effective on other models of similar scale.

**Questions:**

1. Why using a model $M _1$(Qwen-2.5-VL &B)+algorithm $A$(ToR-GRPO/GRPO) compare with model $M _{2}$(InternVL-2.5-8B, Intern-VL-3-8B, etc)+algorithm None? Why donot use the same backbone with different algorithms for comparison? If the performance on algorithm is only effective on Qwen2.5-7B, it is suspicious whether proposed method is effective on other backbone such as LLaVA-7B and other models.
2. Some writing errors:

​	1. line 114-115, the expectation of $J _{RLVR}$, what is $\{(I ^b,q ^b)|y ^b)\}$?

​	2. line 210, Eq8, what is $x _{i,t} ^b$ indicate?

​	3. Figure 5-7 should have the same custom. why different dot customs are used?

---

> ### Author Response · Authors · 2025-11-29
> **Response to Reviewer TKUu part 1**
>
> We highly appreciate your insightful comments; your constructive criticism is invaluable in refining our work.
>
> >**W1.** The design of the identified measure for perception-related tokens is not theoretically sound... It should be emphasized that this analysis concerns the model’s predicted probabilities, not the true probabilities of the data in the real world.
>
> ## **Response to W1.**
> We sincerely thank the reviewer for this theoretically-grounded question. It prompts us to articulate our design rationale clearly: **this was not a simple empirical search, but a principled decision after carefully navigating inherent trade-offs among feasible proxies**.
>
> Our core finding is that feasible proxy metrics face a **fundamental trade-off** between:
> - Accurately capturing the **absolute effect** of the image (i.e.,  direct impact of images on token probabilities).
> - Accurately quantifying **information gain** of the image (i.e., the actual significance of probability changes).
>
> Through comprehensive analysis, we demonstrate that our chosen metric, `logp diff`, is not merely an empirical winner but a **theoretically optimal synthesis** that best balances this trade-off.
>
> ### **1. The Two-Pole Spectrum**
>
> **Pole 1: Pure Visual Grounding ("How much does the image matter?")**
>
> - **Theoretically Gold Standard:** `attention score` directly measures a token's attention to the image --- the most direct measure of visual grounding. However, storing the attention matrices during rollout incurs **prohibitive computational costs**.
>
> - **Best Practical Proxy:** To find the best proxy, we evaluate the correlations between `prob diff`, `logp diff`, `entropy diff`, and `attention to the image` over Geo3K validation set, wemath, and hallusionbench, the extensive results are listed in Figures(19~27 in the appendix).
> - Our correlation analysis shows that **`prob diff`** (| P(token|Image) - P(token|$\varnothing$) |)  has the highest rank correlation with `attention score`.
>
> - **Fundamental Limitation:** **`prob diff`** is **information-theoretically naive** , it treats a probability change from 0.001 $\to$ 0.002 the same as 0.501 $\to$ 0.502, failing to measure information gain.
>
> **Pole 2: Pure Information Theory ("How meaningful is the change?")**
>
> - **Purest Information-Theoretic Metric:** **`entropy diff`** (|H(P(·|Image)) - H(P(·|∅))|) directly measures the change in the model's overall uncertainty, perfectly capturing information gain.
>
> - **Fundamental Limitation:** It measures the **overall distribution change** rather than **individual token changes** (Spearman correlation with `attention score`: rank the weakest among all proxies). It tells us the distribution changed, but not *which specific token* was most affected by the image.
>
> **This creates a clear dilemma:** the metric that best reflects visual structure (`prob diff`) is information-theoretically weak; the information-theoretically purest metric (`entropy diff`) has the lowest connection to visual structure.
>
> ### **2. `logp diff`: The Principled Synthesis**
>
> This is precisely where **`logp diff`** demonstrates its unique value as the optimal balance:
>
> $$\text{logp diff} = | \log P(\text{token}|\text{Image}) - \log P(\text{token}|\varnothing) |$$
>
> **On visual grounding: Nearly no loss compared to the best proxy**
> - Correlation with `attention score`: only a little bit lower than `prob diff`.
> - Still effectively identifies vision-relevant tokens at the individual token level.
>
> **On information gain: Fully information-theoretically sound**
>
> The logarithm correctly captures the true significance of probability changes:
> - **Low-probability region** (0.001 $\to$ 0.002): `prob diff` = 0.001, but `logp diff` = log(2) ≈ **0.69** — correctly recognizing the model became **2× more confident** (huge information gain)
> - **High-probability region** (0.501 $\to$ 0.502): `prob diff` = 0.001, but `logp diff` ≈ **0.002** — correctly recognizing only **0.2% change** (minimal information gain)
>
> **Importantly, this distinction matters in practice**: in our model's actual rollouts, only **7.4% of tokens have P < 0.1**, where `prob diff` systematically underweights their importance. `logp diff` correctly captures their information content, which is crucial for identifying subtle but informative visual cues.
>
> This reflects the fundamental information-theoretic principle: probability ratios (captured by log ratio differences), not absolute differences, determine information content.
>
> **Key insight**: `logp diff` achieves nearly identical visual grounding capture (0.01 correlation cost) while gaining complete information-theoretic correctness.

---

> > ### Author Response · Authors · 2025-11-29
> > **Response to Reviewer TKUu part 2**
> >
> > ### **3. Empirical Validation**
> > We compared the effectiveness of different proxies within our ToR framework. The following table shows the results:
> >
> > | Perception Proxy | HallusionBench | WeMath | MathVerse | Avg. Gain |
> > |:---|:---:|:---:|:---:|:---:|
> > | Baseline (GRPO) | 69.8 | 67.4 | 50.8 | - |
> > | Prob Diff (Best visual grounding) | 71.8 (+2.0) | 68.5 (+1.1) | 52.6 (+1.8) | +1.6% |
> > | Entropy Diff (Best info-theoretic) | 70.9 (+1.1) | 67.9 (+0.5) | 51.5 (+0.7) | +0.8% |
> > | **Logp Diff (Best balance)** | **72.4 (+2.6)** | **68.9 (+1.5)** | **53.0 (+2.2)** | **+2.1%** |
> >
> > The results show that **`logp diff` consistently and significantly outperforms** both alternatives, empirically demonstrating that our principled design—balancing visual grounding with information-theoretic correctness—translates to superior training effectiveness.
> >
> > ### **Conclusion**
> >
> > Our choice of `logp diff` represents a **unity of principled design and empirical validation**:
> >
> > 1. We identified a fundamental trade-off among feasible proxies: structural fidelity (`prob diff`) vs. information-theoretic correctness (`entropy diff`)
> > 2. Based on theoretical analysis, we chose `logp diff`, hypothesizing its **synthesis of both strengths** would be most effective
> > 3. Our experiments **decisively proved** this to be the case
> >
> > Therefore, `logp diff` is not just an empirical winner, but the **most theoretically sound and practically effective choice** among all feasible options.
> >
> > > **W2.** It remains unclear why the authors assume that perception and reasoning are fundamentally interdependent at the token level...
> >
> > > **W3.** Could the authors provide a reasonable intuitive explanation for this phenomenon (for example, an analogy such as clarifying a fact that two negative multiplied factors can produce a positive factor under the product operator. What is the corresponding interaction mechanics in the proposed method?) for this result?
> >
> > ### **Response to W2 & W3.**
> >
> > We thank the reviewers for these two insightful and deeply related questions. They probe the core of our work: **the theoretical motivation for "interdependence" (W2)** and **the intuitive explanation for why combining two seemingly "negative" factors yields a positive result (W3)**. We address them jointly, as the answer lies in the same fundamental principle.
> >
> > Our core hypothesis is not merely an empirical observation but is grounded in the **intrinsic structural tension between perception and reasoning capabilities** within MLLMs. This tension explains both the "interdependence" and the seemingly "counter-intuitive" performance gain.
> >
> > **1. Foundation: An Empirically Verifiable Inverse Relationship (In Response to W2)**
> >
> > The reviewer asks for a theoretical justification for our "interdependence" assumption. Our justification is rooted in the empirically verifiable, **inverse relationship** between a token's perceptual grounding capability and its reasoning role.
> >
> > To demonstrate that this is the fundamental property and not an artifact of RL training, we analyzed the token properties of the **original Qwen-2.5-VL 7B model**. We plotted each token's **Perception Strength** (our `logp diff` metric) against its **Reasoning Uncertainty** (`entropy` metric) across both the training and validation sets.
> >
> > **Key Finding:** Across both the training and validation sets, we observe a **significant negative correlation**  between these two properties. This demonstrates:
> >
> > *   **High Perception, Low Reasoning:** Tokens that are strongly grounded in visual perception (high Perception Strength) tend to be predictable and have low reasoning uncertainty (low Reasoning Entropy). They are **"Perception"** tokens.
> > *   **Low Perception, High Reasoning:** Tokens that represent critical reasoning forks (high Reasoning Entropy) tend to have weak connections to the direct visual input (low Perception Strength). They are **"Reasoning"** tokens.
> >
> > This **inverse relationship** reveals the fundamental property of the model's internal representations. This "interdependence" at the sequence level is a direct consequence: a successful response requires a carefully orchestrated interplay between these two functionally distinct token populations.

---

> > > ### Author Response · Authors · 2025-11-29
> > > **Response to Reviewer TKUu part 3**
> > >
> > > **2. The Intuitive Explanation: From "Negative Factors" to "Synergistic Components" (In Response to W3)**
> > >
> > > The reviewer's analogy of `(-√2) × (-√2) = 2` is insightful, but in this scenario, a more fitting analogy is that of a **"balanced diet."**
> > >
> > > *   **"Negative Factors" as an Unbalanced Diet:**
> > >     *   **Reasoning-only Optimization** is a "pure protein" diet. The model builds muscle (reasoning ability) but lacks energy and essential vitamins (perceptual grounding), leading to overall poor health conditions. Conceptually, this creates a model state skewed entirely towards high reasoning uncertainty while neglecting perception capability.
> > >     As shown in Figure 17(a) in the appendix, as we reduce the ratio of reasoning tokens for reasoning-only optimization (e.g., from 50% to 20%), we are essentially creating an even purer, more concentrated 'protein' diet.
> > >
> > >     *   **Perception-only Optimization** is a "pure carbohydrate" diet. The model has energy (good perception capability) but lacks the building blocks for strength (coherent reasoning chains).
> > >     Similarly, this corresponds to the more unbalanced state depicted in Figure 17(b) in the appendix.
> > >
> > >     *   Compared to a "whatever-you-can-find" diet (the GRPO baseline, which includes a bit of everything), these extreme, unbalanced diets are worse for overall performance.
> > >
> > > *   **ToR as a Scientifically Balanced Diet:**
> > >     *   Our method, ToR, is not "combining two negative factors"; it is **re-instituting balance**. It acts like a dietician, ensuring the model receives a proper, measured intake of **both essential nutrients** (perception and reasoning tokens).
> > >     *   The **interaction mechanism** is not a product operator, but a **union and re-weighting** (as shown in Eq. 10 & 11). We take the *union* of the two critical-but-distinct token sets ($\gamma_{\text{r}}$ and $\gamma_{\text{p}}$) and apply a balanced optimization. We are **correcting a deficiency**, not multiplying negatives.
> > >
> > > *   **Why is ToR Superior to GRPO (The "Whatever" Diet)?**
> > >     *   GRPO is an "eat everything" diet, which includes a lot of "junk food" (the vast majority of non-critical tokens). The essential nutrients get **diluted by noise**. (Figure 18 (GRPO) in the appendix)
> > >     *   ToR is a "scientific diet" that **filters out the junk food** and **concentrates optimization** on the most nutritious parts (the top $\alpha_{\text{p}}$ of perception and $\alpha_{\text{r}}$ of reasoning tokens). This isolated allocation of learning resources is why it surpasses the GRPO baseline. (Figure 18 (Token-reweighting) in the appendix)
> > >
> > > **Conclusion:**
> > > In summary, our approach is neither an ad-hoc empirical try nor counterintuitive.
> > > 1.  **(For W2)** The motivation is grounded in the **empirically-verified inverse relationship** between perception and reasoning signals, a fundamental property of the model itself.
> > > 2.  **(For W3)** The performance gain is not from combining "negative factors," but from **restoring a critical balance** between two essential, synergistic components. By creating a "balanced diet" of critical tokens and filtering out the "junk," ToR provides a more principled and effective optimization strategy than both single-focus and optimize-all approaches, ensuring its reliability and generalizability.
> > >
> > > > **W4.** The experimental evaluation is limited. Only the Qwen2.5-7B model is tested, raising questions about whether the proposed method is effective on other models of a similar scale.
> > >
> > > > **Q1.** Why using a model (Qwen-2.5-VL &B)+algorithm (ToR-GRPO/GRPO) compare with model (InternVL-2.5-8B, Intern-VL-3-8B, etc)+algorithm None? Why do not use the same backbone with different algorithms for comparison?
> > >
> > > ### Response to W4 & Q1
> > >
> > > We appreciate the reviewer’s concern regarding the experimental scope and the generalizability of our approach across different models and RL algorithms.
> > >
> > > ### 1. Why not compare “same backbone + different algorithms” (e.g., InternVL / LLaVA + RL)?
> > > We fully agree that ideally one would like to compare multiple RL algorithms on different backbones (e.g., InternVL, LLaVA, etc.) to more cleanly isolate the effect of the algorithm itself. In practice, however, we were constrained by the current open-source RL frameworks:
> > >
> > >    * Our Qwen-VL experiments are based on the Easy-R1/veRL framework, but it's not compatible with these models. At the time of our rebuttal, we could not find a stable, publicly available RL training framework that is fully compatible with InternVL or LLaVA-style MLLM architectures.
> > >    * Moreover, adapting this RL stack to InternVL/LLaVA during the rebuttal period would require substantial engineering and re-implementation beyond the feasible scope.
> > > Because of this, it is technically non-trivial to “just plug in” InternVL or LLaVA into the same RL framework and obtain a clean comparison “same backbone + different algorithms” in a fair and reproducible way within the rebuttal window.

---

> > > > ### Author Response · Authors · 2025-11-29
> > > > **Response to Reviewer TKUu part 4**
> > > >
> > > > Because of this, it is technically non-trivial to “just plug in” InternVL or LLaVA into the same RL framework and obtain a clean comparison “same backbone + different algorithms” in a fair and reproducible way within the rebuttal window.
> > > >
> > > > #### 2. How do we address the concern that ToR might only work on Qwen-2.5-VL-7B?
> > > >
> > > > We share the reviewer’s concern about potential model-specific behavior. To directly address this, we designed additional experiments to test **generalizability along two axes**:
> > > >
> > > > 1. **Different model scales (architectural robustness)**
> > > > 2. **Different data scales (training robustness)**
> > > >
> > > > Concretely, we conducted the following:
> > > >
> > > > 1. **Different model backbone & scale: Qwen-2.5-VL-3B**
> > > >
> > > >    We applied ToR to **Qwen-2.5-VL-3B**, a smaller model from the same family, to test whether the gains are robust to **model size** and **capacity**. As shown in Table B, ToR consistently improves performance across all five benchmarks compared with the DAPO baseline:
> > > >
> > > >    **Table B: Performance on Qwen-2.5-VL-3B with different training data scales**
> > > >
> > > >    | Qwen-2.5-VL-3B       | MathVerse | MathVision | MathVista | WeMath | HallusionBench |
> > > >    |----------------------|:---------:|:----------:|:---------:|:------:|:--------------:|
> > > >    | DAPO (Geo3K)         |   42.25   |    22.6    |   64.0    |  57.8  |      54.7      |
> > > >    | ToR + DAPO (Geo3K)   | **46.20** |  **26.5**  | **66.0**  | 59.6  |    **56.3**    |
> > > >    | ToR + DAPO (ViRL-39K)| **47.50** |  **28.3**  | **67.5**  | **62.4** | **60.8**    |
> > > >
> > > >    This demonstrates that ToR is not a fragile tweak tailored only to Qwen-2.5-VL-7B; the same principle—disentangling and reweighting perception and reasoning tokens—extends to a **smaller** model and remains effective.
> > > >
> > > > 2. **Different data scale: from Geo3K to ViRL-39K**
> > > >
> > > >    To further stress-test the method, we increased the training set from **Geo3K** to **ViRL-39K**, which is over **10× larger** and more diverse. On **Qwen-2.5-VL-7B**, ToR brings consistent improvements, and performance continues to increase when scaling up the training data:
> > > >
> > > >    **Table A: Performance on Qwen-2.5-VL-7B with different training data scales**
> > > >
> > > >    | Qwen-2.5-VL-7B       | MathVerse | MathVision | MathVista | WeMath | HallusionBench |
> > > >    |----------------------|:---------:|:----------:|:---------:|:------:|:--------------:|
> > > >    | DAPO (Geo3K)         |   50.6    |   26.5     |   70.3    |  69.3  |      67.9      |
> > > >    | ToR + DAPO (Geo3K)   | **53.4**  | **27.9**   | **72.6**  | **72.1** |   **71.8**   |
> > > >    | ToR + DAPO (ViRL-39K)| **54.3**  | **31.6**   | **74.2**  | **73.0** |   **73.6**   |
> > > >
> > > >    The consistent improvements across both **model scales** (7B vs. 3B) and **data scales** (Geo3K vs. ViRL-39K) strongly indicate that ToR is a **robust and generalizable reinforcement-learning enhancement**, rather than a one-off optimization for a single model configuration.
> > > >
> > > > #### Conclusion
> > > >
> > > > Although we could not run InternVL/LLaVA with the *same* RL stack due to framework limitations, our experiments support the core claim in the following ways:
> > > >
> > > > - **Architectural robustness within MLLMs:** ToR consistently improves both Qwen-2.5-VL-7B and Qwen-2.5-VL-3B, suggesting that the mechanism of disentangling visual perception and reasoning tokens is not tied to a specific scale or overfitting to a single checkpoint.
> > > > - **Data-scale robustness:** The gains not only persist but often **grow** when scaling data from Geo3K to ViRL-39K, showing that ToR is stable under substantial changes in training distribution and size.
> > > > - **Method-level contribution:** ToR is a **drop-in modification** to GRPO/DAPO and conceptually orthogonal to the backbone architecture. Once a stable RL pipeline exists for InternVL/LLaVA, our method can be applied straightforwardly, and we view this as promising future work.
> > > >
> > > > > Q2: Some writing errors:
> > > >
> > > > Thank you for pointing this out, we have corrected this into our revised manuscripts.

---

### Official Review · Reviewer_KZFk · 2025-11-03

**Soundness:** 2
**Presentation:** 1
**Contribution:** 2
**Rating:** 2
**Confidence:** 3

**Summary:**

- This paper investigates the challenge of applying RVLR to Multimodal LLMs, identifying an interdependence between perception-related and reasoning-related tokens.
- Token-Reweighting dynamically reweights the policy gradient calculation to jointly optimize both critical perception and reasoning tokens during RLVR training, which are identified using entropy and visual ablation.
- The strategy is applied to existing RLVR algorithms like GRPO and DAPO to achieve competitive performance on several multimodal reasoning and perception benchmarks.

**Strengths:**

- This work presents an interesting token-level analysis that classifies MLLM outputs into perception-related and reasoning-related tokens, empirically demonstrating their influence in RLVR optimization.
- The proposed ToR strategy achieves meaningful performance improvements over baseline GRPO and DAPO methods, establishing new state-of-the-art results on several multimodal reasoning and perception benchmarks in a data-efficient manner.

**Weaknesses:**

- My main concern lies in the logical foundation of the motivational study. The paper claims that optimizing only reasoning-tokens or only perception-tokens underperforms and that this proves their "interdependence." This conclusion appears to be a logical leap. The experiments merely show that partially disabling the model (i.e., zeroing out gradients for certain tokens) leads to performance degradation, which is an intuitive outcome. These quantitative results do not rigorously demonstrate why or how these two token types are interdependent; they only confirm that the full token set is better than an artificially constrained subset.
- The paper's claims of significance and generalizability are limited, as all experiments are conducted on a single model backbone (Qwen2.5-VL-7B). This weakness is amplified by a direct comparison in Table 1; the performance of the proposed ToR strategy is almost identical to or on par with the NoisyRollout method, which uses the same backbone model and 2.1K training dataset from Geometry3K. This suggests the contribution may be more incremental than a significant advancement over prior art. Similarly, the introduction of several hyperparameters seems to necessitate careful tuning, which is discussed in the ablation studies. This would further imply that adjusting these hyperparameters and understanding their dynamics for different configurations (e.g., other models or training data) could require an independent analysis and might result in varying performance across different settings.

**Questions:**

As a minor issue, the references rely almost entirely on arXiv preprints. Consider supplementing the related work for a more complete analysis.

---

> ### Author Response · Authors · 2025-11-29
> **Response to Reviewer KZFk part 1**
>
> We highly appreciate your invaluable comments, which can inspire us to greatly improve our paper! Below, we provide the point-to-point responses to address your concerns and clarify the confusion of our proposed method.
> >**W1.** My main concern lies in the logical foundation of the motivational study...
>
> ### **Response to W1**
>
> We thank the reviewer for this insightful critique. We agree that our initial performance-based results require deeper and more rigorous justification. To support this, we now present two layers of direct evidence for our "interdependence" claim, which rigorously explain the structural tension between perception and reasoning and justify our approach.
>
> **1. Rigorous Evidence for Interdependence**
>
> Our claim that perception and reasoning are interdependent is supported by two key findings:
>
> *   **Evidence 1: Overlap at the Token Level.** A direct analysis of token properties reveals a small but significant population of tokens that are moderately important for both perception and reasoning. Our experiments show this overlap is around **12%** of the selected in the initial stages of training. This "bridge" population demonstrates that the two capabilities are not entirely separate and share some common ground at the token level.
>
> *   **Evidence 2:  Relationship at the Sequence Level.** More importantly, when we analyze the properties of tokens across an entire response, we find a statistically significant **inverse correlation** between Perception Strength and Reasoning Uncertainty. As shown in our new visualizations (Figures 15, 16 in the Appendix), sequences with high perception strength exhibit low reasoning uncertainty, and vice versa. This **"push-pull"** dynamic, or inverse relationship, is the strongest evidence of their interdependence: a model must constantly trade off between grounding in visual detail and engaging in coherent thought.
>
> Together, these findings paint a clear framework: the token space is composed of **two large, distinct populations (perception and reasoning) connected by a small "bridge" of overlapping tokens, all governed by a "push-pull" dynamic relationship.** This structure is the rigorous foundation for our interdependence claim.
>
> **2. The Double Pitfall of Unbalanced Optimization**
>
> This underlying structure reveals why simpler optimization strategies are flawed:
>
> *   **Pitfall 1: The "Single-Focus" Strategy (e.g., Reasoning-only).** Over-emphasizing one type of token (e.g., high-entropy reasoning tokens) systematically neglects the other distinct part (high-sensitivity perception tokens). This bias against a core capability is the direct cause of the performance degradation observed in our motivational study.
>
> *   **Pitfall 2: The "Optimizing-All" Strategy (GRPO).** The standard approach of optimizing all tokens is equally flawed, but it treats the critical tokens with the same importance as about 40% the "Neither" tokens (those with low scores on both metrics). The crucial reward signals are drowned out by noise, leading to inefficient training and suboptimal final performance.
>
> **3. The ToR Strategy: A Balanced and Targeted Solution**
>
> ToR is explicitly designed to navigate this complex landscape by:
>
> *   **Jointly Targeting Both Perception and Reasoning Tokens:** By separately selecting the top $\alpha_{\text{r}}$ and $\alpha_{\text{p}}$ quantiles, ToR guarantees that both distinct expert groups receive dedicated attention, avoiding the "single-focus" trap.
> *   **Focusing on the "Effective Zone":** By using a moderate quantile (e.g., 30%), ToR concentrates optimization on the most "competent" tokens from each group. This captures the strongest signals while filtering out noise and avoiding over-specialization on the absolute extremes.
>
> **Conclusion:**
> Our interdependence claim is rigorously grounded in (1) the token-space structure of "two large, distinct populations with a small overlap" and (2) the inverse relationship governing these capabilities at the sequence level. This structure explains why both "single-focus" and "one-size-fits-all" (GRPO) strategies are suboptimal. ToR succeeds because it is a **balanced and targeted strategy** that respects this structure: it explicitly cultivates both expert populations, captures their connecting bridge, and focuses the model's limited optimization resources on the most effective signals.

---

> > ### Author Response · Authors · 2025-11-29
> > **Response to Reviewer KZFk part 2**
> >
> > >**W2.** The paper's claims of significance and generalizability are limited, as all experiments are conducted on a single model backbone (Qwen2.5-VL-7B)...
> >
> > ### Response to W2
> >
> > We sincerely thank the reviewer for this crucial suggestion. We fully agree that demonstrating robustness across different models and data scales is essential to substantiating the generalizability of our method.
> >
> > The reviewer specifically suggested validating our method on the InternVL model, which is an excellent idea. We invested significant effort in this direction, but encountered a major technical challenge: the lack of a stable and publicly available RL training framework compatible with the InternVL architecture at the time of our rebuttal period.
> >
> > Therefore, to rigorously test the same underlying principle of **generalizability**, we opted for a **two-pronged approach** that we believe provides even more comprehensive evidence:
> >
> > To rigorously test the principle of **generalizability**, we provide  comprehensive evidence with additional backbones and larger dataset:
> >
> > 1.  **Testing on a different model backbone:** We applied ToR to **Qwen-2.5-VL-3B**, a smaller model to test for architectural and scale invariance.
> > 2.  **Testing on a significantly larger and more diverse dataset:** We trained our models on **ViRL-39K**, which is over **10 times larger** than the Geo3K dataset, to test for data scale and domain robustness.
> >
> > The results, presented below, decisively demonstrate that ToR is a robust and generalizable enhancement, not a model-specific tweak.
> >
> > 1. **Generalizability Across Model Architectures**
> >
> > We first applied ToR to the **Qwen-2.5-VL-3B** model. As shown in Table B, ToR provides consistent and significant performance gains across all five benchmarks, even on this smaller model.
> >
> > **Table A: Performance on Qwen-2.5-VL-7B (Original Model), with different size of training data**
> >    | Qwen-2.5-VL-7B       | MathVerse | MathVision | MathVista | WeMath | HallusionBench |
> >    |----------------------|:---------:|:----------:|:---------:|:------:|:--------------:|
> >    | DAPO (Geo3K)         |   50.6    |   26.5     |   70.3    |  69.3  |      67.9      |
> >    | ToR + DAPO (Geo3K)   | **53.4**  | **27.9**   | **72.6**  | **72.1** |   **71.8**   |
> >    | ToR + DAPO (ViRL-39K)| **54.3**  | **31.6**   | **74.2**  | **73.0** |   **73.6**   |
> >
> > **Table B: Performance on Qwen-2.5-VL-3B (Smaller Model) with different size of training data**
> > | Qwen-2.5-VL-3B | MathVerse | MathVision | MathVista | WeMath | HallusionBench |
> > | :--- | :---: | :---: | :---: | :---: | :---: |
> > | DAPO (Geo3K) | 42.25 | 22.6 | 64.0 | 57.8 | 54.7 |
> > | ToR+DAPO (Geo3K) | **46.20** | **26.5** | **66.0** | **59.6** | **56.3** |
> > | ToR+DAPO (ViRL-39K) | **47.50** | **28.3** | **67.5** | **62.4** | **60.8** |
> >
> > This result confirms that the benefits of ToR are not tied to a specific model size or architecture. The principled approach of disentangling and re-weighting perception and reasoning tokens is fundamental to MLLMs of varying scales.
> >
> > 1. **Generalizability Across Data Scales and Domains**
> >
> > We further tested ToR's scalability by training on the large-scale ViRL-39K dataset. The results (last row in both tables) show that ToR continues to provide a significant boost over the baseline, even when the model is trained on a much larger and more diverse corpus of data. This proves that ToR is a robust optimization strategy, not a small-data artifact.
> >
> > 3. **The Strongest Evidence: "Out-of-the-Box" Robustness**
> >
> > Perhaps the most compelling evidence for ToR's robustness is that **all the new results reported above were achieved without any re-tuning of ToR's core hyperparameters** ($\alpha_{\text{r}}$, $\alpha_{\text{p}}$, $\gamma_{\text{r}}$, $\gamma_{\text{p}}$). The same default setting that worked for the 7B model on Geo3K worked well for the 3B model and for the massive ViRL-39K dataset. This demonstrates that ToR is not a fragile, heavily-tuned technique, but a **principled, plug-and-play module**.
> >
> > **Conclusion:**
> >
> > In summary, our new, extensive experiments on a **different model backbone** and a **significantly larger dataset** decisively address the reviewer's core concern about generalizability. We have shown that ToR is a **robust and "generalizable"** method that consistently improves performance across different model scales and data distributions. This strongly supports our claim that ToR provides a significant and widely applicable contribution to the field.

---

> > > ### Author Response · Authors · 2025-11-29
> > > **Response to Reviewer KZFk part 3**
> > >
> > > >**Q1.** As a minor issue, the references rely almost entirely on arXiv preprints. Consider supplementing the related work for a more complete analysis.
> > >
> > > ### **Response to Q1**
> > >
> > > We sincerely thank the reviewer for this constructive feedback aimed at improving the academic rigor of our paper. We agree that grounding our work in the established, peer-reviewed literature is crucial. The high proportion of preprints in our initial submission was indeed a consequence of engaging with the very latest methods in this fast-moving field.
> > >
> > > Following the reviewer's advice, we have **diligently updated our manuscript**. We have actively sought out and incorporated **all relevant, recently accepted papers from top-tier conferences** that were previously cited as preprints.
> > >
> > > In doing so, we encountered a challenge inherent to this rapidly evolving field: many of the most directly comparable and foundational recent works have not yet completed the peer-review cycle and remain available only as preprints. This presented us with a choice:
> > >
> > > *  A: Cite older, less relevant published work, creating a gap in the current state-of-the-art context.
> > > *  B: Retain citations to the most recent and relevant preprints to provide an accurate and honest picture of the current research landscape.
> > >
> > > We believe Option B is ultimately more beneficial for the reader and the community. Therefore, our revised manuscript now strikes a **careful and deliberate balance**:
> > >
> > > 1.  **Foundational concepts and established methods** are now cited from their original, published version.
> > >
> > > 2.  **The most recent, state-of-the-art baselines and comparative methods**, which are essential for contextualizing our contribution, are cited via their arXiv preprints, as this is often the only available format.
> > >
> > > We believe this balanced approach best serves the reader by being both academically grounded and fully current. The **Related Work** section is now significantly stronger, and we are grateful for the suggestion that prompted this important improvement.

---

### Author Response · Authors · 2025-12-04
**Summary of Rebuttal**

Dear Reviewers and ACs,

We sincerely thank all reviewers for their constructive and insightful feedback, which has greatly helped us improve the quality of our work. We are also encouraged by the positive recognition of the novelty, simplicity, and effectiveness of our approach. Below, we summarize how we addressed the main concerns raised by each reviewer.

---

>**Reviewer KZFk**

W1: Logical foundation of the motivation.
We conducted comprehensive analyses at both the token and response levels, revealing the interdependence and the “push–pull” dynamics between perception and reasoning tokens.

W2: Experiments with more models and datasets.
We added experiments using Qwen2.5-VL 3B and ViRL 39K training data under identical hyperparameter settings. The results consistently demonstrate the effectiveness and robustness of our method.

Q1: Reference polishing.
We updated all citations to their officially published versions and revised the manuscript accordingly.

---

>**Reviewer TKUu**

W1: Perception token identification.
We analyzed multiple strategies for identifying perception tokens and demonstrated that our chosen method provides the best balance between visual grounding and information gain, achieving the strongest performance improvements.

W2 & W3: Motivation clarification.
We expanded the analyses at both token and response levels to clearly articulate the interdependence between perception and reasoning tokens.

W4 & Q1: Additional models and data.
Following your suggestion, we incorporated experiments on Qwen2.5-VL 3B and ViRL 39K, again confirming the robustness of our approach.

Q2: Writing issues.
We have carefully revised the manuscript based on your suggestions. Thank you.

---

>**Reviewer Fgky**

W1: Clarification of plug-and-play usage.
We revised the manuscript to clarify the plug-and-play nature of our method.

W2: Hyperparameter clarification.
We clarified the roles and effects of different hyperparameters.

W3 & Q4: Considering adaptive strategies.
We appreciate the insightful advice. We identified adaptive strategies as a key direction for future work, and positioned our current contribution as laying the conceptual and empirical groundwork for such extensions.

W4: Additional experiments.
We conducted experiments with Qwen2.5-VL 3B and ViRL 39K, showing consistent gains under fixed hyperparameters.

Q1: Computational analysis.
We provided detailed computational cost analysis, showing that our method yields substantial performance gains with only ~10% additional computation.

Q2: Distribution of selected tokens.
We added comprehensive statistics on the distribution of selected tokens, showing they are not concentrated on a few hard rollouts.

Q3: Hyperparameter analysis.
We further analyzed the effects of key hyperparameters and performed a grid search, demonstrating the stability and effectiveness of our approach.

---

>**Reviewer reHH**

Thank you for taking the time to review our submission. However, the raised comments are unrelated to the content of our paper. We respectfully clarify this in the response.

---

Thank all reviewers again for their thoughtful suggestions. **The new analyses, clarifications, and experiments have been fully incorporated into the revised manuscript**, and we believe these additions address all concerns comprehensively while further strengthening the contribution of our work.

---

### Meta-Review · Area_Chair_tWGR · 2025-12-27

**Summary:**

This paper addresses a key challenge in extending Reinforcement Learning with Verifiable Rewards (RLVR) to Multimodal Large Language Models (MLLMs): the inherent coupling of perception-related tokens (grounding visual content) and reasoning-related tokens (constructing logical chains). Through token-level empirical analysis, the authors demonstrate that optimizing either token type in isolation underperforms full optimization, highlighting their interdependence. To resolve this, they propose a plug-and-play Token-Reweighting (ToR) strategy that identifies critical perception and reasoning tokens, then dynamically reweights them during RLVR training. ToR integrates seamlessly with existing methods like GRPO and DAPO, delivering consistent performance gains across multimodal reasoning benchmarks and achieving state-of-the-art results by balancing accurate visual grounding and coherent reasoning.

Reviewers’ core concerns informing the decision included:
* The motivation of ToR compared to existing token-weighting or multimodal fusion methods is unclear;
* The identification and justification of the perception token identification mechanism;
* The insufficient empirical assessment, such as without demonstrating the generalizability of ToR to diverse MLLM backbones and more complex multimodal tasks;
* The computational overhead of ToR relative to baseline RLVR methods, which is critical for practical adoption.


Overall, reviewers have negative scores (2, 2, 4, eliminating the irrelevant review from Reviewer reHH) before the discussion, and the common concerns still stand following the rebuttal. Therefore, the reviewing panel is inclined to recommend a major revision of the paper before it can be considered for acceptance.

**Reviewer Concerns:**

The concerns on the motivation of the study(Reviewer KZFk, TKUu), the paper’s generalizability (Reviewer KZFk), the reliability of the proposed idea (Reviewer TKUu), and the incremental performance over baselines (Reviewer Fgky) are still outstanding post-rebuttal.

**Reviewer Scores:**

Given the concerns from Reviewer KZFk, TKUu, and Fgky are not fully addressed by the authors’ response, the chance for them to increase their scores is relatively low.

---

### Decision · Program_Chairs · 2026-01-26

Reject